# Design automation of microfluidic single and double emulsion droplets with machine learning

Ali Lashkaripour ®[1,2] ✉, David P. McIntyre ®[3,4], Suzanne G. K. Calhoun ®[5], Karl Krauth[2], Douglas M. Densmore ®[3,4,6] & Polly M. Fordyce ®[1,2,7,8] ✉

Droplet microfluidics enables kHz screening of picoliter samples at a fraction of the cost of other high-throughput approaches. However, generating stable droplets with desired characteristics typically requires labor-intensive empirical optimization of device designs and flow conditions that limit adoption to specialist labs. Here, we compile a comprehensive droplet dataset and use it to train machine learning models capable of accurately predicting device geometries and flow conditions required to generate stable aqueous-in-oil and oil-in-aqueous single and double emulsions from 15 to 250 μm at rates up to 12000 Hz for different fluids commonly used in life sciences. Blind predictions by our models for as-yet-unseen fluids, geometries, and device materials yield accurate results, establishing their generalizability. Finally, we generate an easy-to-use design automation tool that yield droplets within 3 μm (<8%) of the desired diameter, facilitating tailored droplet-based platforms and accelerating their utility in life sciences.

Droplet microfluidics enables massively parallel miniaturized assays by stably dispersing nanoliter to picoliter samples of a liquid (the dispersed fluid) within an immiscible carrier liquid (the continuous fluid)[1]. Single emulsion (SE) water-in-oil or oil-in-water droplet systems have unlocked opportunities in single-cell omics[2–4], directed evolution[5,6], chemical synthesis[7], and drug and antibody discovery[8,9]. Double emulsion (DE) droplets commonly consist of an aqueous core wrapped in an oil shell that is dispersed in an aqueous outer continuous fluid[10] and have been used for controlled drug delivery[11,12], production of microparticles with core-shell structures[13,14], and in the food and cosmetics industries[15,16]. Due to their aqueous outer fluid and high stability, DEs can also be sorted using commercial fluorescence-activated cell sorting (FACS) machines, enabling off-the-shelf screening in droplet microfluidics at kHz throughput[17–20].

Despite the benefits of droplet microfluidics, adoption of this technology in life sciences has been limited primarily to specialized

groups or commercially available products with limited functionality (e.g., 10× Genomics Chromium machines[21,22]). A major limiting factor is that droplet stability, size, and generation rates dictate downstream assay performance but are difficult to predict. The effective concentration rates of species in droplet assays scales inversely with the 3rd power of droplet diameter, and single-cell encapsulation and the efficiency of FACS sorting are highly size-dependent[17,23]. Precise control over the generation rate is similarly crucial for the development of integrated multi-component microfluidic platforms[24].

Droplets are most commonly made using flow-focusing geometries that yield highly monodisperse droplets over a wide range of diameters and generation rates and require low continuous-to-dispersed flow rate ratios[25–28]. However, the complex and highly nonlinear dynamics of multi-phase flows and the large number of effective parameters in flow-focusing geometries have made it difficult to establish an analytical solution or a generalizable scaling formula that

[1]Department of Bioengineering, Stanford University, Stanford, CA, USA. [2]Department of Genetics, Stanford University, Stanford, CA, USA. [3]Department of Biomedical Engineering, Boston University, Boston, MA, USA. [4]Biological Design Center, Boston University, Boston, MA, USA. [5]Department of Chemical Engineering, Stanford University, Stanford, CA, USA. [6]Department of Electrical & Computer Engineering, Boston University, Boston, MA, USA. [7]Chan-Zuckerberg Biohub, San Francisco, CA, USA. [8]Sarafan ChEM-H Institute, Stanford University, Stanford, CA, USA. ✉e-mail: alilp@stanford.edu; pfordyce@stanford.edu

can accurately predict droplet diameter and rate across a broad range of flow conditions and fluid properties[29,30]. These limitations are exacerbated in life sciences, where biological assays require buffers with varying properties (e.g., interfacial tension and viscosity) that significantly impact the resultant droplet diameter and generation rate[31,32]. As a result, generating droplets with desired properties typically requires multiple resource-intensive design iterations and empirical tests[33,34], and this process becomes even more challenging when integrating other components upstream or downstream of a droplet generator[24,35]. Thus, a tool capable of accurately predicting device geometries and flow rates required to create droplets with desired properties could dramatically simplify device design, facilitate multi-component devices, and facilitate broader adoption of these platforms in life sciences[34].

Machine learning models trained on experimental data were recently demonstrated to enable accurate prediction of SE droplet generation performance[36]. However, previously proposed models only account for variations in flow rates and device geometries[37,38] or surfactants[39]. As a result, previous models offer limited utility in life science applications.

Here, we leverage machine learning and a comprehensive experimental dataset including both SE and DE droplets comprised of many different fluids to train models that accurately predict droplet diameter and generation rate across a diverse range of fluid properties, geometries, flow rates, and device surface properties. In addition, we demonstrate that our models generalize to additional device geometries, fluids, and materials by experimentally validating "blind" predictions using additional device geometries, fluid compositions, and

materials. Finally, we integrate these predictive models with an automated search algorithm to create a design automation tool for SE and DE droplets. This online and open-source tool, called DAFD 3.0 (Design Automation of Fluid Dynamics), can return the necessary design and flow rates to achieve the user-specified diameter and rate for different fluids, while also predicting other characteristics such as performance range and stability (Fig. 1).

## Results

### Comprehensive droplet generation dataset

To generate a comprehensive dataset detailing the impacts of device designs, flow rates, and fluid properties on droplet diameters and generation rates, we curated and combined two previously generated SE and DE experimental datasets[32,37]. This comprehensive dataset includes 46 different polydimethylsiloxane (PDMS) and polycarbonate device designs (43 SE and 3 DE generators with 49 flow-focusing geometries combined), 8 different dispersed fluids, and 6 different continuous fluids for generating aqueous-in-oil and oil-in-aqueous droplets of 15–250 μm in diameter at rates of 5–12,000 Hz (Fig. 2).

We previously generated aqueous-in-oil (DI water and mineral oil) SEs using 43 devices and multiple flow rate combinations[37]. This dataset varied the orifice width from 75 to 175 μm and systematically explored the remaining geometric parameters according to the orifice width (Fig. 2a). The devices were then tested at a range of capillary numbers and flow rate ratios (see Methods for definitions) and yielded droplets of 25–250 μm at 5–500 Hz in the dripping regime (474 datapoints total). To improve generalizability, we first converted the orifice

**Fig. 1 | Pipeline for collating data and training models to enable performance prediction and design automation of SE and DE droplet generation.**
**a** Composition of datasets exploring effects of geometry (orifice width ($W_{or}$), continuous inlet width (CIW), dispersed inlet width (DIW), channel height (H), and outlet channel width (OCW)), fluid properties, and flow rates on (i.) SE and (ii.) DE droplet generation collated to yield a final (iii.) comprehensive dataset with 868 entries. The combined dataset includes 8 dispersed fluids, 6 continuous fluids, and 46 devices that yield aqueous-in-oil and oil-in-aqueous droplets with diameters

from 15 to 250 μm at rates of 5–12,000 Hz. **b** Schematic of model training to predict: (1) droplet diameter based on device geometry, fluid properties, and flow rates, and (2) droplet generation rates based on predicted diameters and conservation of mass (see Methods). **c** Predictive models were integrated with a custom search algorithm to convert user-specified desired droplet characteristics to an optional device design and flow rates. This open-source software tool, DAFD 3.0, is available at: dafdcad.org. Source Data are provided as a Source Data file.

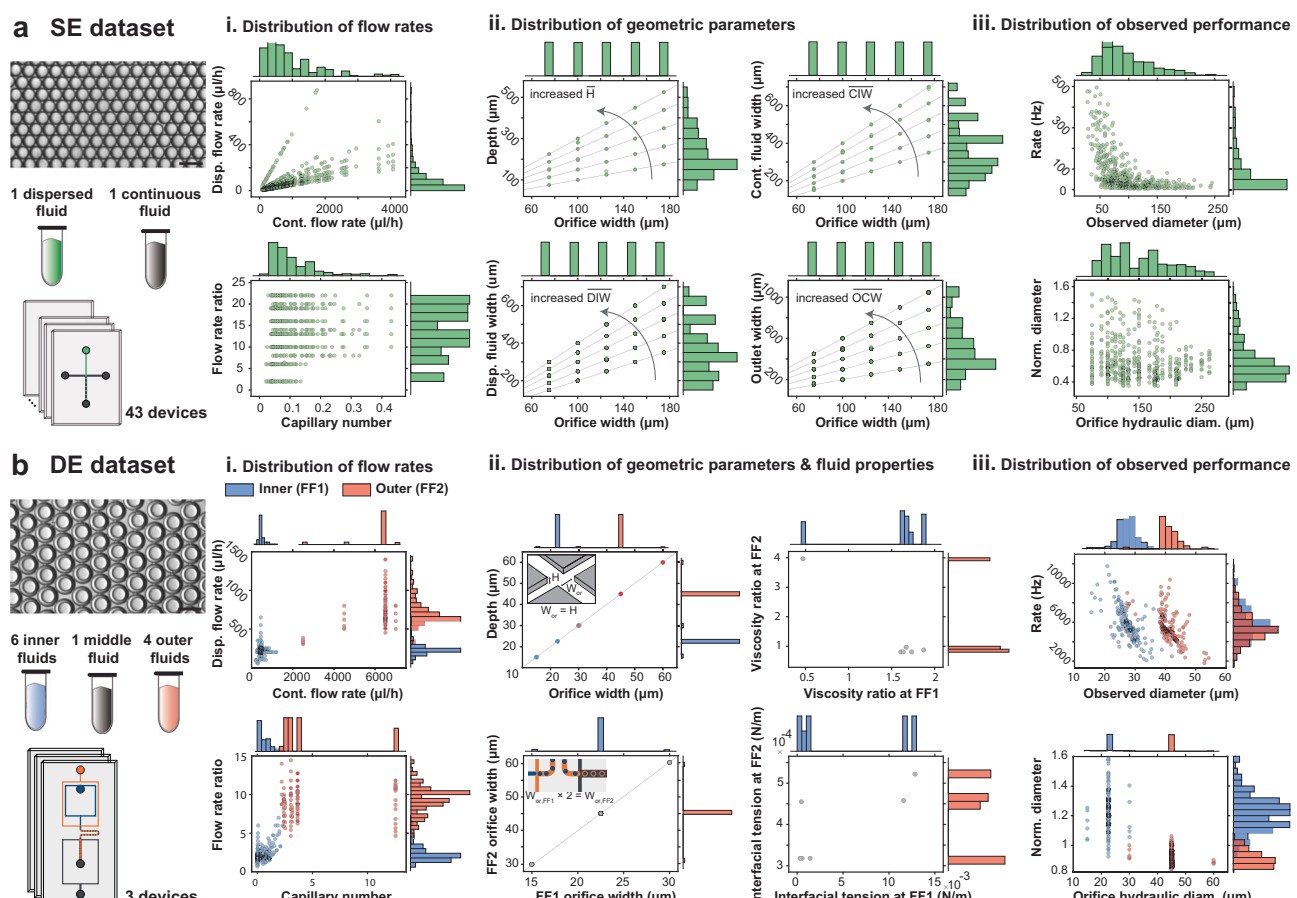

**Fig. 2 | The comprehensive dataset includes SE and DE droplets produced using a wide variety of device geometries, fluid properties, and flow rates. a** The SE dataset includes 43 polycarbonate devices that varied orifice width, channel depth, outlet width, and continuous and dispersed fluid inlet widths; each device was used over multiple flow rate ratios and capillary numbers to generate SEs with diameters of 25–250 μm at rates of 5–500 Hz. A representative image of SEs for a single experiment is shown. **b** The DE dataset includes 3 PDMS devices (each with two flow-focusers) and 6 inner, 1 middle, and 4 outer fluids. Devices were tested at different flow rates of inner, middle, and outer fluids (different capillary numbers and flow rate ratios for each junction). The orifice widths of the PDMS devices were 15, 22.5, and 30 μm at the first junction, with an aspect ratio of 1, while the orifice size at the 2nd junction was twice the size of the 1st junction. Several biologically relevant fluids with different viscosities and interfacial tensions were used to generate droplets with diameters of 15 (inner) to 54.2 (outer) μm at rates of 1800–11,800 Hz. A representative image of DEs for a single experiment is shown. Scale bars represent 50 μm. Source Data are provided as a Source Data file.

dimensions of each device to hydraulic diameter ($D_h$):

$$D_h = \frac{2 \cdot W_{or} \cdot H}{W_{or} + H}, \quad (1)$$

where $W_{or}$ is orifice width and $H$ is channel height. We then computed normalized droplet diameter $\bar{D}$ produced by each device:

$$\bar{D} = \frac{D}{D_h}, \quad (2)$$

where $D$ is the observed droplet diameter. The diverse range of flow rates (Fig. 2a.i) and device design parameters (Fig. 2a.ii) resulted in droplets with normalized diameters ranging from 0.35 to 1.5 (Fig. 2a.iii).

We also previously generated aqueous-oil-aqueous DEs using 3 PDMS devices with different geometries using multiple flow rates (197 datapoints total). These experiments used several biologically relevant fluids with applications in cell culture, cell lysis, and molecular biology (e.g., PCR, NGS, and scATAC-Seq) including 6 different inner, 1 middle, and 4 outer fluids (Table 1)[32]. The 3 devices had orifice widths of 15, 22.5, and 30 μm at flow-focuser 1 (FF1) and 30, 45, and 60 μm at flow-

focuser 2 (FF2), respectively, with a normalized channel depth (i.e., aspect ratio) of 1. A serpentine flow resistor between FF1 and FF2 largely decoupled the generation dynamics of FF1 and FF2 and enabled easy surface modification; the orifice at FF2 was twice as wide and deep as the orifice at FF1 to ensure a mostly dripping-dripping mode of DE generation for higher chances of stable DE generation and better control over inner and outer diameters[32,35,40]. The resultant droplet diameters ranged from 15.5 to 54.2 μm and generation rates varied from 1800 to 11,800 Hz. To create a generalizable predictive model for droplet generation and to effectively model DE generation, we considered DE generation as two independent SE generation events at FF1 and FF2, with FF1 generating aqueous-in-oil SEs and FF2 generating oil-in-aqueous SEs. We also normalized DE inner and outer diameters using the hydraulic diameters at FF1 and FF2, respectively (i.e., the orifice at which droplets are generated). Normalized inner diameters varied from 0.92 to 1.6 (15.5–42.1 μm) and normalized outer diameters ranged from 0.84 to 1.06 (27.4–54.2 μm), as shown in Fig. 2b.

We then curated and combined the SE and DE datasets by using standardized definitions of capillary number and geometric parameters (see Methods) to create a comprehensive dataset of microfluidic droplet generation that covers a diverse design space of capillary numbers, flow rate ratios (Supplementary Fig. 1), geometries,

**Table 1 | Eleven different fluid combinations in the dataset make it possible to investigate the effects of fluid properties on droplet generation**

| | Dispersed fluid | | Continuous fluid | | Interaction | |
|---|---|---|---|---|---|---|
| | Fluid | Viscosity (mPa · s) | Fluid | Viscosity (mPa · s) | Interfacial tension (mN/m) | Device material |
| | *Fluids included in the comprehensive dataset* | | | | | |
| 1. | M9 bacterial media | 0.861 | HFE 7500 oil + 2.2% ionic Krytox | 1.61 | 12.84 | PDMS |
| 2. | M9 bacterial media + 25 mM glucose | 0.967 | HFE 7500 oil + 2.2% ionic Krytox | 1.61 | 11.60 | PDMS |
| 3. | PBS | 0.931 | HFE 7500 oil + 2.2% ionic Krytox | 1.61 | 0.543 | PDMS |
| 4. | PBS + 1% Tween-20 | 0.988 | HFE 7500 oil + 2.2% ionic Krytox | 1.61 | 0.319 | PDMS |
| 5. | PBS + 0.9% NP40 | 1.003 | HFE 7500 oil + 2.2% ionic Krytox | 1.61 | 1.41 | PDMS |
| 6. | PBS + 10% PEG 6000 mw | 3.431 | HFE 7500 oil + 2.2% ionic Krytox | 1.61 | 0.461 | PDMS |
| 7. | DI water | 1.001 | NF350 mineral oil + 5% Span 80 | 57.2 | 5.0 | PC[a] |
| 8. | HFE 7500 oil + 2.2% ionic Krytox | 1.61 | PBS+1% Tween-20 + 2% Pluronic F68 | 1.303 | 0.318 | PDMS[b] |
| 9. | HFE 7500 oil + 2.2% ionic Krytox | 1.61 | M9 salts + 2% Pluronic F68 | 1.412 | 0.522 | PDMS[b] |
| 10. | HFE 7500 oil + 2.2% ionic Krytox | 1.61 | M9 salts + 25 mM glucose + 2% Pluronic F68 | 1.563 | 0.458 | PDMS[b] |
| 11. | HFE 7500 oil + 2.2% ionic Krytox | 1.61 | PBS + 10% PEG 6000 mw + 2% Pluronic F68 | 6.395 | 0.455 | PDMS[b] |
| | *Unseen fluids for assessing the generalizability of models* | | | | | |
| 12. | DMEM complete cell media + 16% Optiprep | 1.25 | 2% dSurf HFE 7500 oil | 1.61 | 9.74 | PDMS |
| 13. | RPMI 1640 complete cell media + 20% Optiprep + 0.1% Pluronic F127 | 1.36 | 2% dSurf HFE 7500 oil | 1.61 | 6.16 | PDMS |
| 14. | 2% dSurf HFE 7500 oil | 1.61 | RPMI 1640 complete cell media + 5% Pluronic F127 | 2.82 | 4.61 | PDMS[b] |
| 15. | Surfactant-free HFE 7500 oil | 1.31 | Trimethylolpropane trimethacrylate (TMPTMA) | 42 | 3.10 | PDMS[b] |
| 16. | Trimethylolpropane trimethacrylate (TMPTMA) | 42 | 50% glycerol in water | 6 | 15.3 | PDMS[b] |
| 17. | 52% w/w glycerol in water (surfactant-free) | 7 | Silicone oil | 4.6 | 29 | Glass |
| 18. | 52% glycerol in water + 50 mM dodecyl-trimethylammonium bromide | 7 | Silicone oil | 4.6 | 10 | Glass |
| 19. | 52% glycerol in water + 5 mM hexadecyl-trimethylammonium bromide | 7 | Silicone oil | 4.6 | 7.3 | Glass |

These fluids are commonly used across different life science applications. The viscosity of the dispersed fluids varied from 0.86 to 3.4 mPa · s and the viscosity of the continuous fluids varied from 1.61 to 57.2 mPa · s. The interfacial tension between the dispersed and continuous fluids ranged from 0.318 to 12.84 mN/m. In addition, generated data and previously published data on droplet generation using 8 unseen fluid combinations were used to assess the generalizability of models to unseen fluids.
[a]Polycarbonate.
[b]PDMS made hydrophilic with plasma treatment.

fluid properties, and output performance (Table 2). As orifice length minimally impacts droplet generation in the dripping regime, we did not consider it as a design parameter[41]. This enabled us to model flow-focusing geometries with and without an orifice constriction (a simple straight channel, where orifice length cannot be clearly defined). As droplet diameter and generation rate are independent of surface properties or device material as long as surface properties favor droplet generation (i.e., continuous fluid completely wetting the channel)[42–44], we did not consider device material or surface properties as a design parameter. This dataset is the largest experimental dataset available for microfluidic droplet generation in the dripping regime and includes aqueous-in-oil and oil-in-aqueous droplets, different biologically relevant fluids, and various device materials (Supplementary Table 1).

**Droplet diameter and generation rate prediction**

We trained scaling law, neural network, and boosted decision tree models to predict SE and DE droplet diameters and generation rates. While scaling laws (i.e., empirically fitted scaling formulas) are simple and have been traditionally used for this task, they are often inaccurate or fail to generalize to unseen fluids and size scales[37,39]. We therefore also trained machine learning models and compared

their accuracy and generalizability to scaling laws. To improve generalizability, we made all design parameters dimensionless when possible. This involved using capillary number, viscosity ratio, and flow rate ratio to account for fluid properties (i.e., viscosity and interfacial tension) and flow rates. We also normalized all geometric parameters (channel depth, dispersed and continuous inlet widths, and outlet channel width) by the orifice width (except for the orifice width itself) (Fig. 3a). In all models, we first predicted normalized droplet diameter based on input parameters and then used the hydraulic diameter of the orifice to calculate an actual droplet diameter:

$$D_p = \overline{D_p} \cdot D_h , \qquad (3)$$

where $D_p$ is predicted droplet diameter, $\overline{D_p}$ is predicted normalized diameter, and $D_h$ is the hydraulic diameter of the orifice.

To evaluate the accuracy of models and prevent overfitting, we randomly split the comprehensive dataset into a training set (80%) and a testing set (20%) for 15 different training sessions and calculated the average performance of each model against the test set. For each model, we first predicted droplet diameters and then used these values to calculate predicted generation rates based on dispersed fluid flow

**Table 2 | The comprehensive dataset includes 868 datapoints on single and double emulsion generation with different fluids**

| Parameter | Unit | Lower bound | Upper bound | Unique values |
|---|---|---|---|---|
| **Output performance** | | | | |
| Droplet diameter | μm | 15.5 | 245.1 | 825 |
| Generation rate | Hz | 5 | 11,774 | 838 |
| Droplet diameter normalized by hydraulic diameter | N.A. | 0.35 | 1.60 | 833 |
| **Fluid properties** | | | | |
| Dispersed fluid viscosity | mPa·s | 0.861 | 3.431 | 8 |
| Continuous fluid viscosity | mPa·s | 1.303 | 57.2 | 6 |
| Interfacial tension | mN·m$^{-1}$ | 0.318 | 12.84 | 11 |
| Viscosity ratio (continuous/dispersed) | N.A. | 0.47 | 57.2 | 11 |
| **Geometric parameters** | | | | |
| Orifice width | μm | 15 | 175 | 10 |
| Normalized channel depth | N.A. | 1 | 3 | 13 |
| Normalized continuous fluid inlet width | N.A. | 1 | 4 | 15 |
| Normalized dispersed fluid inlet width | N.A. | 1 | 4 | 13 |
| Normalized outlet channel width | N.A. | 1 | 6 | 15 |
| **Flow parameters** | | | | |
| Flow rate ratio (continuous/dispersed) | N.A. | 0.69 | 22 | 187 |
| Capillary number | N.A. | 0.014 | 9.399 | 206 |

This dataset includes both aqueous-in-oil and oil-in-aqueous droplets with a broad range of output performance. This is achieved by varying effective parameters in flow-focusing droplet generation including device geometry, fluid properties, and flow rates.
Entries in bold represent a category of design parameters.

rate and conservation of mass (assuming stable droplet generation with a uniform diameter):

$$Q_d = F \cdot V_d \qquad (4)$$

$$F = \frac{6 \cdot Q_d}{\pi \cdot D^3}. \qquad (5)$$

Here, $Q_d$ is the dispersed fluid flow rate, $F$ is the generation rate, $V_d$ is the droplet volume, and $D$ is the droplet diameter. To predict the generation rate at FF2 of DE generators based on outer diameter, we set the flow rate of the dispersed fluid to the total flow rate of inner and middle fluids (as is required to satisfy conservation of mass).

Fitting several previously published scaling laws[45–50] to the comprehensive dataset yielded predictions with a mean absolute percentage error (MAPE) range of 17.7–47.6% for diameter predictions and 58.7–3023% for rate predictions (Supplementary Note 1, Supplementary Table 2 and Supplementary Figs. 2–12). Among these models, the Liu et al. scaling law showed the best accuracy and used flow rate ratio, viscosity ratio, and capillary number as inputs (Fig. 3b)[45]. These inputs may not always affect diameter independently and the impact of flow rate ratio can vary from low to high capillary numbers[41]. Therefore, we also proposed a scaling law that accounts for some level of parameter dependence. This scaling law was able to predict diameter and generation rate with a MAPE of 13.6% and 46.9%, respectively (Fig. 3c). Including additional parameters as inputs either prevented finding a solution or reduced accuracy.

Next, we trained a neural network that takes capillary number, flow rate ratio, and five geometric parameters (orifice width, normalized channel depth, normalized outlet width, normalized dispersed fluid inlet width, and normalized continuous fluid inlet width) as inputs and predicts normalized droplet diameter. We did not include viscosity ratio as an input for the neural network as it resulted in a slightly lower accuracy when predicting blind data with as-yet-unseen fluids and geometries (discussed in Generalizability to unseen geometries and fluids section), despite achieving a slightly higher accuracy for the comprehensive dataset

(Supplementary Note 2 and Supplementary Table 3). We chose a wide and shallow network structure, with 2 hidden layers of 512 and 16 nodes, respectively, which is more suitable for small datasets compared to deep and narrow structures (i.e., more hidden layers with fewer nodes)[51]. The trained neural network significantly outperformed the scaling laws over 15 randomized sessions, with MAPE of 7.4% for diameter and 22.6% for generation rate (Fig. 3d, see Supplementary Fig. 13 for 14 additional training sessions).

We then trained boosted decision trees to predict normalized droplet diameters using viscosity ratio, capillary number, flow rate ratio, and the five geometric parameters as inputs. Across 15 randomized training sessions, boosted decision trees showed an MAPE of 5.4% for predicting diameter and 16.6% for generation rate (Fig. 3e, see Supplementary Fig. 14 for 14 additional training sessions). Overall, boosted decision trees (closely followed by the neural network) enabled the most accurate performance prediction in flow-focusing aqueous-in-oil and oil-in-aqueous droplet generation across different fluids with diameters of 15–250 μm at rates of 5–12,000 Hz; models showed higher accuracy for predicting the inner diameter of DEs compared to their outer diameter (Supplementary Fig. 15). Other statistical metrics including coefficient of determination ($R^2$), mean absolute error (MAE), and root mean square error (RMSE) also demonstrate the significantly higher accuracy of machine learning models compared to scaling laws (Table 3).

Machine learning models show even greater improvements for predicting generation rate. Literature scaling laws resulted in a negative $R^2$ (i.e., predictions were worse than just predicting the mean outcome for all outcomes) and a MAE of 1367 Hz (MAPE of 58.7%) for predicting generation rate, compared to $R^2 = 0.98$ and an MAE of 220 Hz (MAPE of 16.6%) for boosted decision trees and $R^2 = 0.97$ and MAE of 260 Hz (MAPE of 22.6%) for neural network. For both machine learning models, the MAPE for generation rate was approximately three times the MAPE for diameter. This is mathematically expected according to the conservation of mass. As the generation rate inversely scales with the 3rd power of diameter, assuming a relatively small error in diameter prediction and using a Taylor series expansion yields a 3-fold larger MAPE for rate

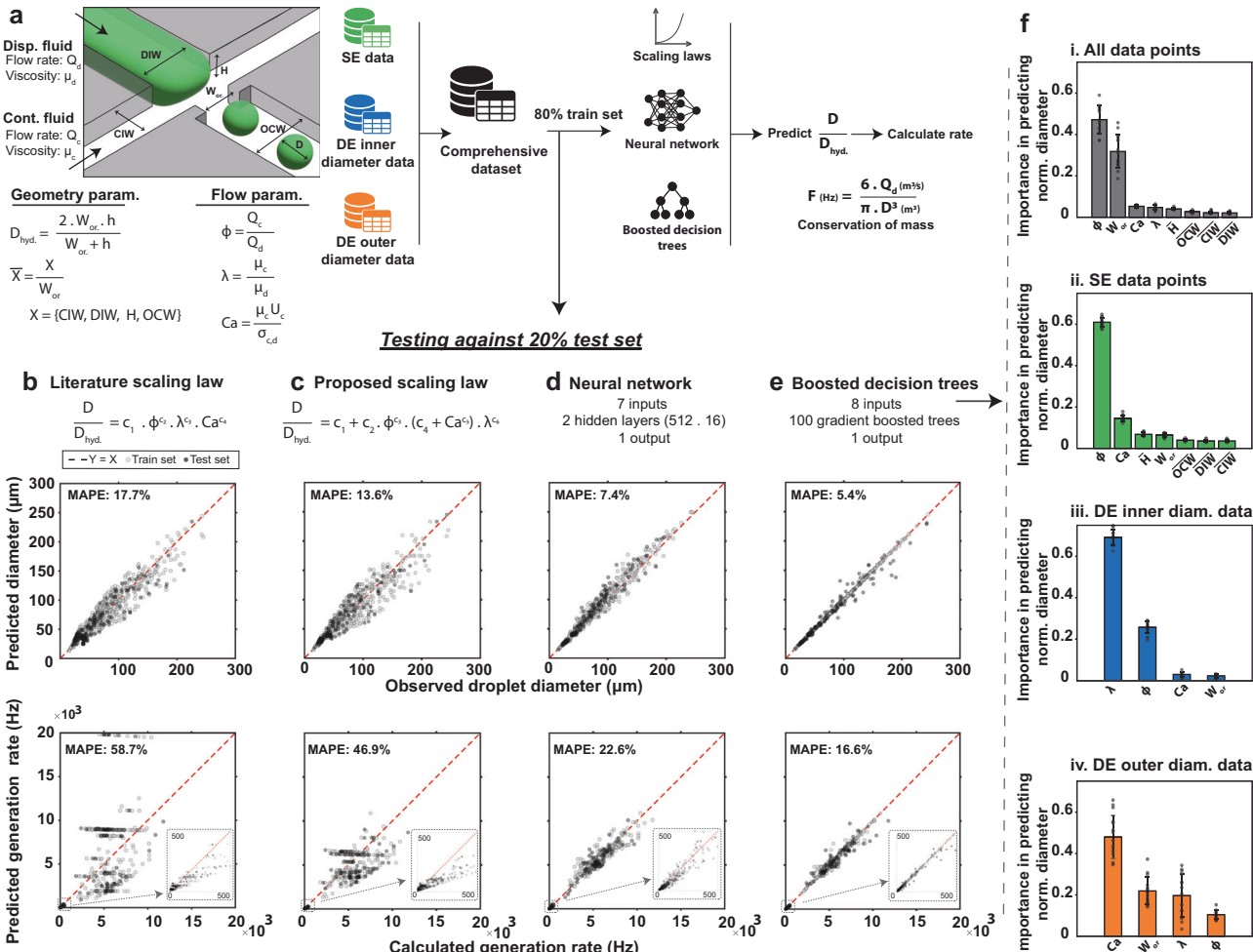

**Fig. 3 | Boosted decision trees and neural networks accurately predict SE and DE droplet diameters and generation rates. a** To develop generalizable models, we converted fluid and flow properties and geometric parameters to dimensionless inputs (flow rate ratio (Φ), viscosity ratio (λ), capillary number (Ca), and normalized geometric parameters ($\overline{X}$), see Methods) and split the comprehensive dataset into 80% train and 20% test sets across 15 randomized sessions. In each case, we trained and compared performance of **b** a previously published scaling law[45], **c** a proposed scaling law, **d** a neural network and **e** boosted decision trees. MAPEs for predicting rate were approximately 3 times the MAPEs for diameter, as expected from conservation of mass. Red dashed line indicates the 1:1 line, each gray marker indicates model-predicted values for datapoints included either within the training set (light gray) or the test set (dark gray) of a single representative model. **f** Relative importance of different parameters in predicting droplet diameters with boosted decision trees; bars represent the average significance and error bar total length represents two standard deviations across 15 random training sessions. Source Data are provided as a Source Data file.

## Table 3 | Performance prediction accuracy for each model

| Parameter | MAPE | $R^2$ | MAE | RMSE |
|---|---|---|---|---|
| Droplet diameter prediction | | | | |
| Boosted decision trees | $5.38 \pm 0.34\%$ | $0.96 \pm 0.00$ | $4.62 \pm 0.41\ \mu m$ | $8.29 \pm 0.73\ \mu m$ |
| Neural network | $7.45 \pm 0.42\%$ | $0.95 \pm 0.01$ | $5.98 \pm 0.38\ \mu m$ | $9.88 \pm 0.49\ \mu m$ |
| Proposed scaling law | $13.65 \pm 0.68\%$ | $0.88 \pm 0.02$ | $9.89 \pm 0.62\ \mu m$ | $14.96 \pm 1.14\ \mu m$ |
| Literature scaling law | $17.67 \pm 0.45\%$ | $0.89 \pm 0.01$ | $10.85 \pm 0.38\ \mu m$ | $14.51 \pm 0.80\ \mu m$ |
| Generation rate prediction | | | | |
| Boosted decision trees | $16.58 \pm 1.05\%$ | $0.98 \pm 0.00$ | $219.7 \pm 12.7\ Hz$ | $452.5 \pm 19.5\ Hz$ |
| Neural network | $22.59 \pm 1.4\%$ | $0.97 \pm 0.00$ | $260.1 \pm 14.3\ Hz$ | $499.9 \pm 22.9\ Hz$ |
| Proposed scaling law | $46.86 \pm 4.60\%$ | $0.83 \pm 0.03$ | $676.9 \pm 80.7\ Hz$ | $1182.6 \pm 90.1\ Hz$ |
| Literature scaling law | $58.68 \pm 3.09\%$ | $-0.02 \pm 0.24$ | $1367.5 \pm 590.9\ Hz$ | $2774.7 \pm 582.1\ Hz$ |

Metrics are reported for a 20% test set, using the average ± the standard deviation for 15 different randomized training and testing sessions.
*MAPE* Mean absolute percentage error, $R^2$ coefficient of determination, *MAE* mean absolute error, *RMSE* root mean square error.

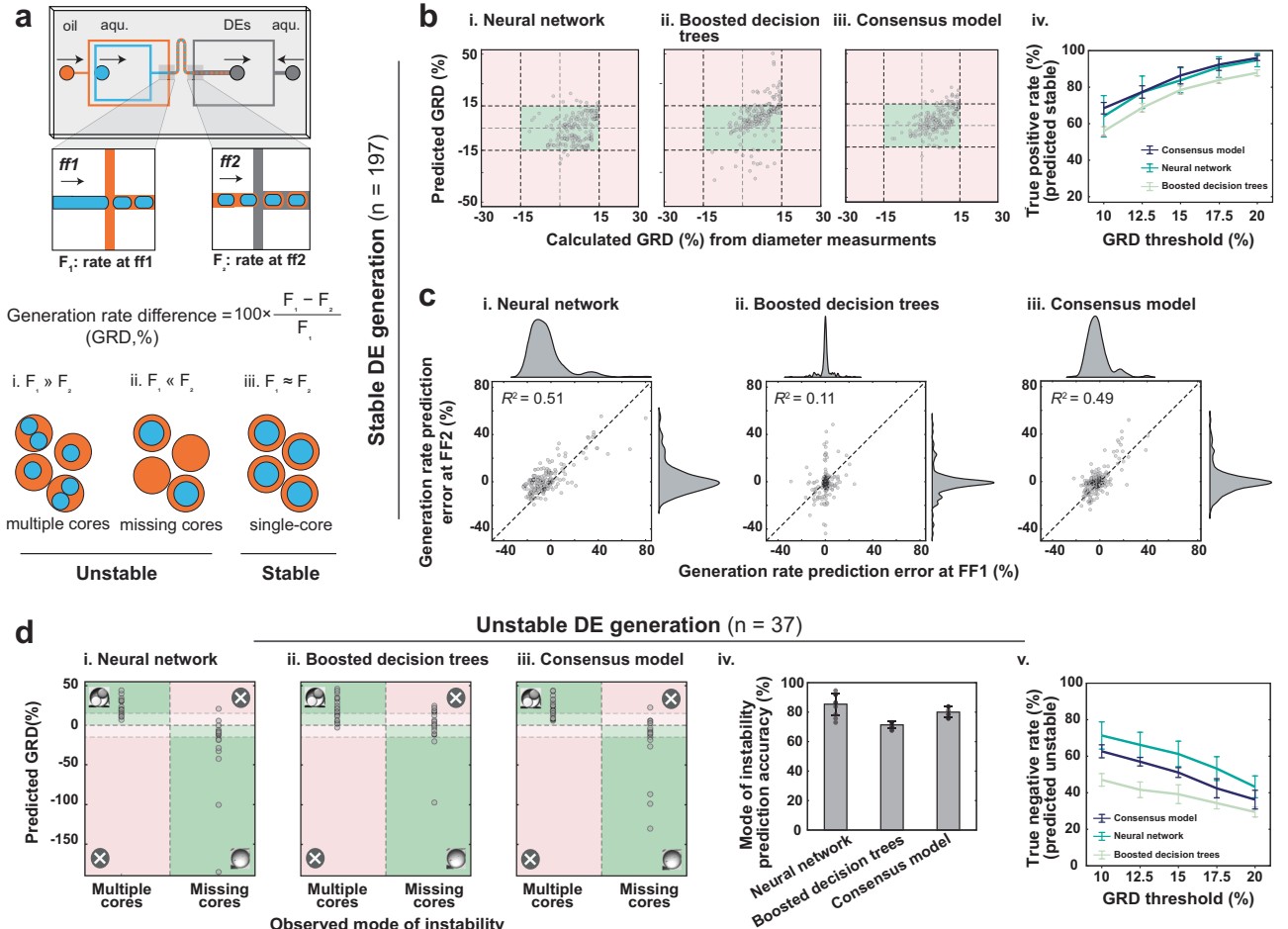

**Fig. 4 | Models' performance in predicting stable single-core and unstable DE droplet generation regimes. a** DE generation was modeled as two events of droplet generation at FF1 (aqueous-in-oil) and FF2 (oil-in-aqueous); the threshold for unstable DE generation (multiple core or missing core) was set to a generation rate difference (GRD) of 15%. **b** Comparison of predicted and calculated GRD at FF1 and FF2 for (i.) the neural network, (ii.) boosted decision trees, and (iii.) consensus model over the 197 stable datapoints. Green boxes indicate regions with predicted GRD <15% and experimentally stable DEs; predictions are shown for a single representative model. (iv.) The effect of GRD threshold on the true positive rate of stable DE generation predictions (i.e., correctly predicted to be stable) for different models; values represent average accuracies and error bars indicate two standard deviations over 10 randomized training sessions. **c** Comparisons between errors in

model-predicted generation rates at FF1 and FF2. Markers show comparisons for a single representative model and dashed line indicates 1:1 line. **d** Comparisons between observed mode of instability vs. predicted GRD for (i.) the neural network, (ii.) boosted decision trees, and (iii.) the consensus model. Correct predictions appear in green shaded areas, incorrect predictions appear in red shaded areas, and GRDs predicted to lead to stable droplets are indicated by lighter shading. (iv.) Plots quantifying accuracy in predicting the mode of instability and (v.) the true negative rate (i.e., percentage of unstable DE generation data that were correctly predicted to be unstable) for different GRD thresholds. Plot values indicate average accuracy of predictions and error bar's total length indicates two standard deviations over 10 randomized training sessions. Source Data are provided as a Source Data file.

prediction (Supplementary Note 3)[37]. Scaling laws deviate from this rule likely because their error in predicting diameter is not sufficiently small to neglect higher order approximations in Taylor series expansion.

Boosted decision trees are interpretable and can reveal the relative significance of design parameters for a dataset (Methods: Parameter significance study). To determine key parameters in different scenarios of droplet generation, we trained and evaluated decision trees on different subsets of the comprehensive dataset. Flow rate ratio, orifice width, capillary number, and viscosity ratio were most important for predicting normalized droplet diameter in the comprehensive dataset (Fig. 3f.i). For aqueous-in-oil SE droplets, flow rate ratio remained the most important, followed by capillary number (Fig. 3f.ii). For DE droplets, normalized inner diameters were mostly determined by viscosity ratio and flow ratio (Fig. 3f.iii) while normalized outer diameters were affected by all parameters, with capillary number being the most significant (Fig. 3f.iv).

### Prediction of stable and unstable DE generation

Producing stable single-core DE droplets requires that generation rates at FF1 and FF2 be matched. If the rate at FF1 exceeds that of FF2, some DEs end up with multiple cores; conversely, if the rate at FF1 is lower than that at FF2, some droplets do not contain a core (Fig. 4a). As generation rates depend critically on device geometry and fluid properties, identifying conditions required to generate stable single-core DEs for unseen reagent combinations is typically a time-consuming process involving several design iterations and flow rate optimizations for inner, middle, and outer fluids. Here, we tested if machine learning models trained to predict diameter and generation rate could streamline this process by classifying whether particular device geometry and flow rate combinations lead to stable or unstable DE generation (i.e., single-core DEs or multiple/missing core DEs). To accomplish this, we leverage the fact that our models can consider DE generation as a combination of two independent droplet generation events (i.e., generation of aqueous-in-oil droplets and oil-in-aqueous

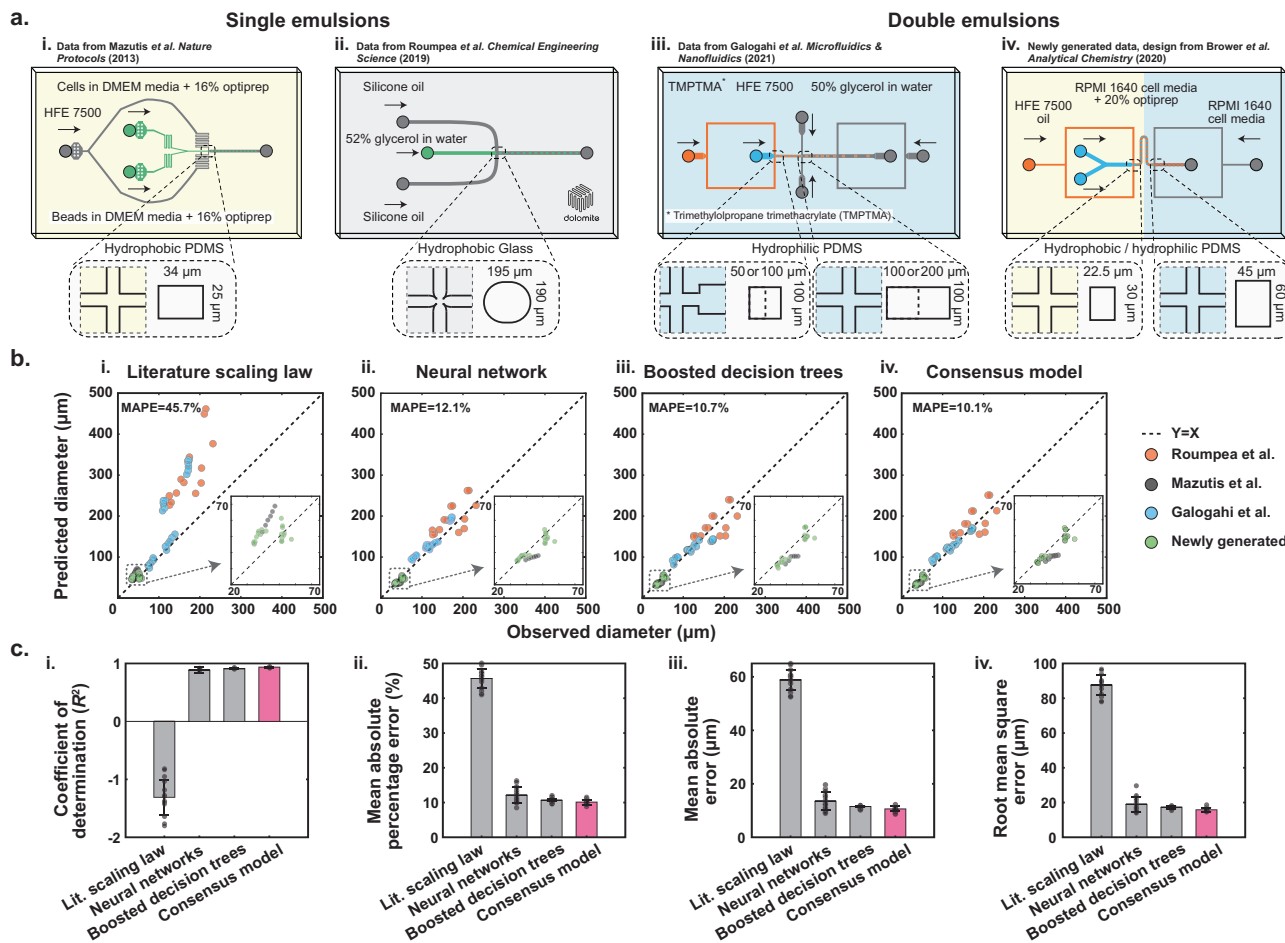

**Fig. 5 | Generalization of machine learning models to fluids, geometries, and device materials not included within the comprehensive dataset. a** References for previously published and additionally generated experimental data and schematics showing device geometries, surface properties, and fluid compositions used to generate SE (two left devices) and DE droplets (two right devices). **b** Comparison between observed diameters and those predicted by a previously proposed scaling law and 3 different trained models for a single representative training session.

Dashed lines indicate 1:1 identity line, and annotation denotes the average mean absolute percentage error over 15 randomized training sessions. **c** Quality metrics model performance in predicting droplet diameter. Bars indicate average performance across 15 randomized training sessions; error bars' total length denotes two standard deviations. Magenta color highlights the top-performing model. Source Data are provided as a Source Data file.

droplets) and the physical knowledge that generating stable DEs requires minimizing the difference in generation rates at FF1 and FF2.

We assessed our machine learning models by (1) predicting the stability of 197 datapoints in the comprehensive dataset that resulted in stable DE generation, (2) generating 37 additional datapoints on unstable DE generation using 5 different fluid combinations, and (3) predicting the instability and mode of instability for the additional datapoints. For the 197 stable datapoints, we observed a maximum generation rate difference (GRD) of 15% between the experimentally calculated generation rates at FF1 and FF2. As non-single-core DEs result when FF1 and FF2 generation rates are mismatched, it is somewhat surprising that stable DEs can be produced with mismatches of this magnitude. This discrepancy likely stems from small inaccuracies in experimentally measured diameters, which scale by a power of 3 when calculating generation rates. We therefore classified any set of conditions with a predicted absolute GRD of <15% as yielding stable droplets:

$$GRD\,(\%) = 100 \cdot \left| \frac{F_1 - F_2}{F_1} \right|. \qquad (6)$$

Here, $F_1$ and $F_2$ are generation rates at FF1 and FF2, respectively, calculated using inner and outer diameters and conservation of mass.

Using these criteria, the neural network correctly predicted conditions that generate stable, single-core DEs for 83.8% of stable DE datapoints over 10 randomized training sessions (Fig. 4b.i). Despite predicting diameters and generation rates more accurately than neural networks, boosted decision trees correctly classified conditions as producing stable DEs for only 78.5% of cases (Fig. 4b.ii); in the remaining cases, conditions that generated stable droplets were predicted to be unstable. This performance difference likely stemmed from differing degrees of correlation between model-predicted rate errors at FF1 and FF2 ($R^2 = 0.51$ and $R^2 = 0.11$ for the neural network and boosted decision trees, respectively, Fig. 4c.i, ii). To take advantage of the high accuracy of boosted decision trees in predicting generation rates and the high accuracy of the neural network in predicting DE stability, we developed a consensus model that averages predictions of each model (i.e., mean of diameters). This consensus model correctly predicted stability for 86.4% of datapoints while also reducing generation rate prediction errors (Fig. 4b.iii, c.iii). Increasing the GRD threshold for DE stability yielded higher true positive rates in predicting stable DE generation for all models (Fig. 4b.iv).

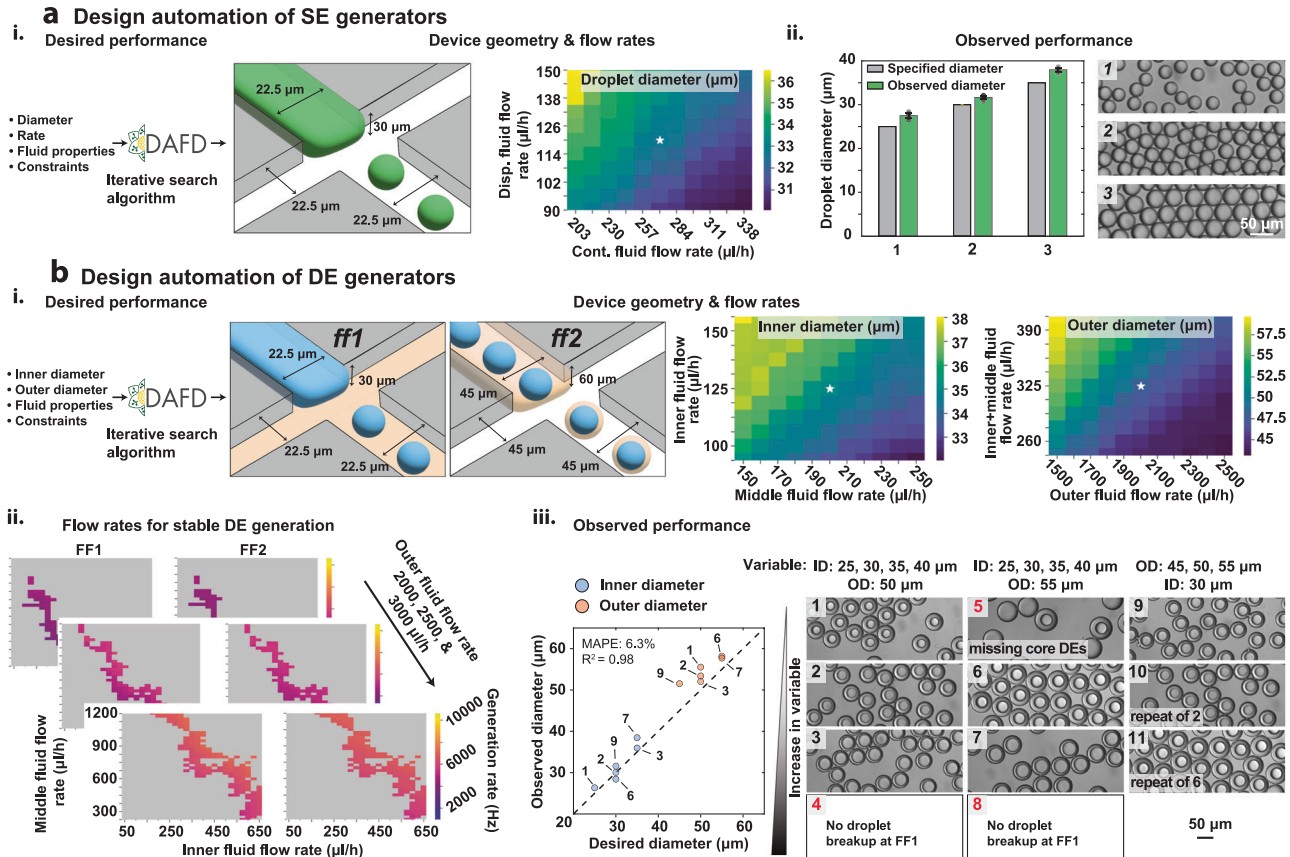

**Fig. 6 | Trained machine learning models and custom search algorithms enable design automation of SE and DE droplet generation. a** Design automation of SE droplet generation. (i.) DAFD 3.0 takes user-specified diameter, rate, fluid properties, and optional constraints as inputs and returns the necessary geometry and flow rates required to generate the desired droplets. (ii.) DAFD-predicted and measured droplet diameters after specifying desired SE diameters of 25, 30, and 35 μm for an unseen fluid combination (left) and representative images of generated droplets (right). Measured droplets differed from specified droplets by a MAE of 2.3 μm (MAPE of 7.9%). Bars indicate average measured diameter and error bars' total length represents two standard deviations in diameter across 10 measured droplets. **b** Design automation of DE droplet generation. (i.) DAFD 3.0 also converts user-specified DE inner and outer diameters to the necessary geometries and flow rates required to generate them. (ii.) DAFD-predicted generation rates as a function of middle, inner, and outer flow rates are used to predict generation rate differences (GRDs) between FF1 and FF2 to identify likely stable (GRD <5%) and unstable (GRD >5%, gray shaded areas) regimes. (iii.) Comparison between observed and DAFD-specified DE inner (blue) and outer (orange) diameters for an unseen fluid combination and 9 different flow rates (left); images show representative DE droplets generated under each condition (right). For stable droplets, observed inner and outer diameters differed from those specified by an MAE of 2.7 μm (MAPE of 6.3%). Experiments were carried out once. Two sets of droplets generated 5 min apart were analyzed to report the mean diameter. Scale bars represent 50 μm. Source data are provided as a Source Data file.

Next, we tested if these models (trained on the entire comprehensive dataset instead of a randomly selected 80%) could predict unstable DE generation for previously unseen data despite only being trained on conditions that lead to stable DE generation. Specifically, we generated 37 additional datapoints using 5 different fluids (Supplementary Fig. 16) to make DE droplets with multiple or missing cores, and tested if these models correctly predicted unstable DE generation and its mode of instability. While machine learning models accurately predicted the mode of instability (i.e., GRD >0 : multiple cores or GRD <0 : missing cores) for unstable DEs (neural networks, boosted decision trees, and the consensus model-predicted modes of instability correctly in 85.4%, 71.4%, and 80.0% of cases across 10 random training sessions, respectively), they were less able to predict if the degree of instability precluded stable DE generation (neural networks, boosted decision trees, and the consensus model-predicted absolute GRD > 15% for 61.3%, 39.2%, and 51.1% of unstable generation cases across 10 random training sessions, respectively) (Fig. 4d.i–iv). Decreasing the GRD threshold for DE instability increased the true negative rate of all models in predicting unstable DE generation (Fig. 4d.v). Varying the GRD threshold for stability/instability can help maximize chances of finding a flow rate combination that yield stable DEs or minimize chances of operating at flow conditions that yield unstable DEs (Supplementary Fig. 17). The prediction performance of this classification could likely be improved in the future either by training a regime classifier on a dataset that includes an equal representation of stable and unstable DE generation cases.

## Machine learning models can generalize to previously unseen geometries and fluids

Training models that accurately generalize to unseen design parameters and data sources is a common challenge in developing machine learning models[52–54]. Here, we directly tested the ability of each model (literature scaling model, neural network, boosted decision trees, and consensus model) to generalize by: (1) using each model to predict droplet diameter and generation rate for as-yet-unseen fluids, device geometries, and device materials, (2) comparing model predictions with previously published diameter and generation rate measurements, and (3) comparing blind model predictions with additionally generated measurements using an unseen device geometry and as-yet-unseen fluid combinations. This evaluation of the accuracy of "blind" model predictions provides a stringent test of the degree to which each model can generalize.

First, we compared model predictions with previously published but as-yet-unseen datasets including aqueous-in-oil SEs produced using PDMS[26] and glass[55] devices and oil-polymer-aqueous DEs droplets produced using PDMS devices[56]. In the first SE dataset, droplets of complete DMEM mammalian cell media with added 16% optiprep were generated using a PDMS device and HFE 7500 oil containing 1.5% fluorinated surfactant for single-cell analysis (Fig. 5a.i)[26]. In the second SE dataset, droplets of 52% glycerol in water with three different surfactant combinations were generated using a commercial glass device (Dolomite) and silicone oil (Fig. 5a.ii)[55]. In the DE dataset[56], core-shell structures were formed using HFE 7500, trimethylolpropane trimethacrylate (TMPTMA), and 50% glycerol in DI water as inner, middle, and outer fluids, respectively (Fig. 5a.iii). These three datasets include a total of 44 datapoints of droplet inner and outer diameters within the diameter range of our models (15–250 μm).

Next, we fabricated a DE generator device based on a previously published design[17] with unseen channel geometries (a normalized channel depth of 1.33 instead of 1 and two aqueous inlets instead of a single inlet) and used it to generate DEs with as-yet-unseen fluids suitable for mammalian cell encapsulation (complete RPMI 1640 cell media with 20% optiprep and 0.1% pluronic F127 for inner fluid, dSurf HFE 7500 for middle fluid, and complete RPMI 1640 with 5% pluronic F127 for outer fluid). We generated DEs using 10 different flow rate combinations (a total of 20 inner and outer diameters), yielding droplets with inner diameters of 34.1–41.7 μm and outer diameters of 51.2–57.5 μm. Finally, after measuring the interfacial tension for each fluid interface (required to calculate capillary number if not previously reported, see Methods), we used the pre-trained models to predict droplet diameters and generation rates of all datapoints (64 total) and directly compared model predictions to experimental data.

As seen previously, predicted droplet diameters were least accurate for the literature scaling law with MAPE of 45.7% (56.5% rate, Supplementary Fig. 18) and a negative coefficient of determination. The machine learning models were all consistently more accurate, with the consensus model slightly outperforming others in terms of MAPE (10.1% for diameter and 29.6% for rate, see Supplementary Fig. 18) and coefficient of determination ($R^2$ of 0.94 and 0.95 for diameter and rate, respectively) averaged over 15 randomized training sessions (Fig. 5b,c). This ability to accurately predict data with as-yet-unseen fluids, geometries (in case of DE generators demonstrated for devices with FF2 orifice widths that are 2-fold the width at FF1), and device materials demonstrates an ability to generalize, likely due to the diversity of the comprehensive dataset in terms of geometries, fluid properties, and flow rates, the use of dimensionless inputs and output, and $L2$ regularization during model training[57].

## Design automation of SE and DE droplets

The ability to automate the design of devices for producing droplets with desired diameters and generation rates can dramatically reduce time spent fabricating, testing, and optimizing microfluidic devices. We previously developed an online open-source tool (DAFD, for Design Automation of Fluid Dynamics) that converted user-specified droplet diameters and rates into a microfluidic design and flow rates that delivered the desired performance[37]. However, the previous version of this tool was limited to only aqueous-in-oil SEs, a single simple fluid combination (DI water and mineral oil), large polycarbonate devices (orifice width >75 μm), and a maximum generation rate of 500 Hz. Here, we present an open-source-tool, DAFD 3.0, that leverages the consensus model (i.e., average of neural network and boosted decision trees) and automated search algorithms to design devices with different materials capable of producing aqueous-in-oil and oil-in-aqueous SE and DE droplets using a wide variety of different fluids. This tool supports device orifice widths of 15–175 μm and droplets of 15–250 μm in diameter produced at rates of 5–12,000 Hz.

For SE design automation, DAFD 3.0 takes the desired diameter and rate alongside the viscosities and interfacial tension of dispersed and continuous fluids as inputs and provides the necessary device geometry and flow rates (while allowing for optional design constraints; Supplementary Fig. 19 and Methods). To test DAFD 3.0's accuracy and reliability for SEs, we specified that we wanted to produce SEs with diameters of 25, 30, and 35 μm using an as-yet-unseen fluid combination (RPMI 1640 complete cell media with added 20% optiprep and 0.1% pluronic F127 as dispersed fluid and dSurf HFE 7500 as the continuous fluid) and constrained the possible geometry to require the same pre-fabricated DE generator device used to assess model generalizability in Fig. 5a.iv. We then used the model-suggested parameters to generate SEs with the DE generator by blocking the outer fluid inlet and flowing dispersed and continuous fluids through FF1. Introducing fluids using DAFD-suggested flow rates (Supplementary Table 4) yielded SEs of 27.5, 31.6, and 37.9 μm in diameter, very close to model predictions with an overall MAE of 2.36 μm and MAPE of 7.94% (Fig. 6a).

For DE design automation, our tool takes desired inner and outer diameters and fluid properties (viscosities and interfacial tensions) of three fluids as inputs and predicts DE inner and outer diameters generated using either six different default designs or a user-specified geometry (if a suitable solution can be found). For DE generation, DAFD 3.0 requires that: (1) the total flow rate at FF1 (i.e., the inner plus middle fluid flow rates) equals the flow rate of dispersed fluid at FF2 to uphold conservation of mass, and (2) that the GRD at FF1 and FF2 to be <5% to ensure stable DE generation (Fig. 6b.i, ii and Supplementary Fig. 17). DAFD 3.0 then ranks potential solutions based on their average deviation from the desired inner and outer diameters and returns a single best set of flow rates (Supplementary Fig. 20 and Methods).

To test DAFD 3.0's accuracy and reliability for DEs, we specified that we wanted: (1) to generate DEs with inner and outer diameters of 25–40 and 45–55 μm, respectively, using the same as-yet-unseen fluids as used above for SEs (RPMI 1640 complete cell media with added 20% optiprep and 0.1% pluronic F127 as inner fluid and dSurf HFE 7500 as the middle fluid) and an outer fluid of 5% pluronic F127 and (2) to constrain the design to the same DE device geometry used above (Fig. 5a.iv). We then used the 9 suggested flow rate combinations to generate DEs and quantified the resultant droplet diameters (Supplementary Table 5). Consistent with prior observations that our models cannot perfectly predict whether flow combinations yield stable single-core droplets, 3/9 conditions at the extremes of inner droplet diameter did not yield single-core DEs (highlighted in red in Fig. 6b.iii). The flow rates suggested to create DEs with the largest inner diameter (40 μm inner diameter and either 50 or 55 μm outer diameter) led to no droplet formation at FF1, while the flow rates suggested to create DEs with a 25 μm inner diameter and a 55 μm outer diameter led to many DEs missing a core. Among stable datapoints, DAFD 3.0 was highly accurate, generating DEs that were different from target diameters by an MAE of 2.70 μm (MAPE of 6.3%). The accuracy for inner diameter (MAE of 1.5 μm and MAPE of 4.8%) was higher than outer diameter (MAE of 3.9 μm and MAPE of 7.9%), potentially due to the bias of training data toward aqueous-in-oil droplets and the minimal yet non-zero dependence of outer diameter on the inner diameter (Fig. 6b.iii). Finally, after validating DAFD 3.0's ability in performance prediction and design automation of SE and DE droplet generation with as-yet-unseen fluids, we trained the machine learning models on all datapoints generated and used here (including previously published and additionally generated data) and integrated the updated models with the online tool.

## Discussion

Here, we establish that machine learning models can enable accurate and generalizable prediction of droplet diameters and generation

rates based on device geometries, fluid properties, and flow rates for aqueous-in-oil and oil-in-aqueous SE and DE droplets (shown for DE generators with FF2 widths 2-fold greater than the width at FF1). These models cover droplet diameters of 15–250 µm at rates of up to 12,000 Hz and take a broad range of input design parameters, including three orders of magnitude variation in capillary number, two orders of magnitude variation in viscosity ratio, and more than one order of magnitude variation in flow rate ratio and microfluidic channel size. This represents a notable improvement over previous models for single-fluid aqueous-in-oil SEs that could only account for variations in flow rates and geometry for generation rates of up to 500 Hz[37] or surfactants for a single geometry (Supplementary Table 1)[39]. In all cases, the trained neural network and boosted decision trees detailed here outperform previously published scaling laws and machine learning models in terms of accuracy and parameter range[37,39,45–50]. Nonetheless, a consensus model based on both the neural network and boosted decision trees resulted in better generalizability to as-yet-unseen fluids and geometries.

Our models account for variations in fluid properties, flow rates, and geometry by considering seven dimensionless inputs (capillary number, flow rate ratio, viscosity ratio, normalized channel depth, normalized dispersed fluid inlet width, normalized continuous fluid inlet width, and normalized outlet width) and only the orifice width as an input with units. Dimensionless inputs and a dimensionless output (droplet diameter normalized by hydraulic diameter of orifice) and a diverse training set enable generalizability to additionally created and previously published data that the models were not trained on. Moreover, our models' accuracy in predicting stable and unstable DE generation demonstrates their utility for both aqueous-in-oil and oil-in-aqueous droplets and validates our simplifying assumption that dripping-dripping DE generation can be modeled as two independent droplet generation events with only minimal loss in accuracy.

Predictive models can be integrated with custom search algorithms to create design automation tools that return device geometries and flow rates optimal for producing droplets with user-specified characteristics. Our tool also enables rapid performance characterization of existing droplet generators. For instance, all flow rate combinations that result in stable DE generation and the diameters of the resulting droplets can be quickly predicted. Using the consensus model that averages predictions of the neural network and boosted decision trees, we created an online and open-source software tool to eliminate the need for design iterations when developing SE and DE generators (DAFD 3.0). DAFD 3.0 requires the viscosities and interfacial tension of fluids used for droplet generation, which can be readily measured using standard rheological and in-situ techniques[32,58] or approximated using properties of similar fluids and the Good and Girifalco model for interfacial tension (see Methods)[59]. DAFD 3.0's diverse training set and an improved training approach allowed it to significantly outperform DAFD 1.0[37] in terms of accuracy, range of design parameters, and generalizability to as-yet-unseen fluids and geometries (Supplementary Note 4, Supplementary Table 7, and Supplementary Figs. 21 and 22).

The generalizable predictive power of our models results in part from training on two independently created datasets for microfluidic droplet generation (Supplementary Fig. 22). We therefore envision future versions of DAFD to benefit from publicly available datasets to achieve higher accuracy and account for a broader range of parameters. Future repositories for microfluidic data in addition to repositories for device designs such as Metafluidics[60] would greatly benefit community-driven design automation efforts. Future integration of our tool with other computer-aided design tools for microfluidics[61,62] and real-time dynamic control schemes[63,64] would enable automated conversion of high-level user specifications to fabrication-ready designs that robustly deliver the desired performance.

Sophisticated high-throughput microfluidic operations require multiple integrated components to function optimally in tandem. As the number of on-chip components increases, the possible design space grows exponentially, making designing and optimizing such platforms challenging[33]. A predictive understanding of microfluidic components allows the realization of additional functionalities. For example, here we establish that a predictive understanding of SE generators allows performance prediction and design automation of DE generators (a two-component device). Similarly, we expect that a predictive understanding of droplet generators in conjunction with deterministic lateral displacement arrays, inertial focusers, pico-injectors, and cell and droplet sorters will dramatically accelerate the design of high-throughput screening platforms.

## Methods
### Dimensionless numbers and flow rate calculations
Droplet generation can be considered as a competition between viscous forces exerted by the continuous fluid and cohesive forces within the dispersed fluid[30]. As a result, the capillary number (given by the ratio of viscous forces to interfacial tension) is commonly used to describe the characteristics of droplet generation. In flow-focusing droplet generation, capillary number can be defined as given in Eq. (7):

$$\mathrm{Ca.} = \frac{\mu_c \cdot U_c}{\sigma_{c,d}} , \qquad (7)$$

where $\mu_c$ is the dynamic viscosity of the continuous fluid, $U_c$ is the characteristic velocity of the continuous fluid through the orifice (flow rate of the continuous fluid divided by the cross-sectional area of orifice), and $\sigma_{c,d}$ is the interfacial tension between the continuous and dispersed fluids.

The flow rate of the continuous fluid can therefore be calculated using capillary number, fluid properties, and device geometry as given in Eq. (8):

$$Q_c = \frac{\mathrm{Ca.} \cdot \sigma_{c,d} \cdot W_{or} \cdot H}{\mu_c} , \qquad (8)$$

where $Q_c$ is the flow rate of the continuous fluid, $W_{or}$ is the orifice width, and $H$ is channel depth.

The flow rate of the dispersed fluid can be determined from the flow rate of the continuous fluid and the flow rate ratio as follows:

$$Q_d = \frac{Q_c}{\Phi} , \qquad (9)$$

where $Q_d$ is the dispersed fluid flow rate, $Q_c$ is the flow rate of the continuous fluid, and $\Phi$ is the flow rate ratio.

Some of the predictive models developed here also take viscosity ratio as an input, defined using Eq. (10):

$$\lambda = \frac{\mu_c}{\mu_d} , \qquad (10)$$

where $\mu_c$ is the dynamic viscosity of the continuous fluid and $\mu_d$ is the dynamic viscosity of the dispersed fluid.

To improve the generalizability of our models to different size scales, we converted the geometric parameters of a flow-focusing device to dimensionless numbers by normalizing them by the orifice width (except for orifice width itself);

$$\overline{X} = \frac{X}{W_{or}} . \qquad (11)$$

Here, $\overline{X}$ is the dimensionless (normalized) geometric parameter, and $X$ can be channel height ($H$), dispersed inlet width (DIW), continuous inlet width (CIW), or outlet channel width (OCW). Orifice length was not considered as a design parameter in our models due to its negligible effect on droplet size and rate in the dripping regime. This also enables us to model flow-focusing geometries with unclear orifice lengths (e.g., when the outlet channel has the same width as the orifice width such that the orifice length could equally be considered to be 0 or equal to the length of outlet channel).

## Measurement of fluid properties

Standard pendant drop tensiometry with drop shape analysis was used to measure the interfacial tension between different pairs of dispersed and continuous fluids, as reported in Table 1 and as previously described[32]. Since the density of HFE 7500 oil is greater than the density of inner and outer fluids used in this study, pendant oil droplets were suspended within the inner or outer aqueous fluids to measure the interfacial tension. A metal capillary nozzle (27 gauge) was used to suspend oil droplets within 5 mL of inner or outer fluid. A custom MATLAB code was used to analyze the oil droplet shape and calculate interfacial tension, as previously established[65,66]. Briefly, shape analysis was conducted when the oil droplet was as stable as possible. Since droplets were observed at equilibrium, cohesive forces (interfacial tension) and gravitational deformation are balanced and the simplified Young-Laplace equation can be equated to hydrostatic pressure and solved to estimate interfacial tension. All reported interfacial tension measurements were the average of 3–6 analyzed drops.

Here, we measured the dynamic viscosity of fluids using a commercial rotational cone and plate rheometer, as previously described[32]. Briefly, a 2° cone at 20 °C was used to conduct a logarithmic flow sweep across a broad range of shear rates (2.86479–2864.79 Hz). The average viscosity in the linear regime was reported as the shear rate-independent viscosity of the fluid.

In case of limited access to measurement equipment, the interfacial tension between two liquids can also be approximated with the Good and Girifalco model[59], using the surface tension of the two liquids and their molar volume:

$$\sigma_{c,d} = \sigma_c + \sigma_d - 2\Phi(\sigma_c \cdot \sigma_d)^{1/2}. \tag{12}$$

Here, $\sigma_{c,d}$ is the interfacial tension, $\sigma_c$ and $\sigma_d$ are the surface tension of continuous and dispersed fluids, and $\Phi$ is the interaction coefficient, defined as:

$$\Phi = \frac{4(V_c \cdot V_d)^{1/2}}{V_c^{1/2} + V_d^{1/2}}, \tag{13}$$

where, $V_c$ and $V_d$ are the molar volume of continuous and dispersed fluids, respectively.

## Device fabrication

SE droplet generators were rapidly prototyped using a low-cost desktop micromill (Bantam Tools) to ablate microfluidic channels with the smallest dimension of 75 μm out of a polycarbonate substrate, as previously described[67]. DE droplet generators were fabricated through standard photolithography followed by soft-lithography as previously described[32]. Briefly, a silicon wafer with two different heights (the height at flow-focuser 2 is double the height at flow-focuser 1) was created using 2-layer SU8 deposition and standard photolithography[17]. To cast PDMS devices from this master mold, we poured a 1:5 ratio of PDMS on the wafer, degassed, and cured for 15 min at 80 °C. Inlets and outlets were punched using a 1 mm biopsy punch (Robbins Instruments) and then this featured layer was placed on a blank slab of 1:10

PDMS (that was cured for 15 min at 80 °C) and baked for 48 h at 80 °C to bond the device to the blank slab (via off-ratio bonding) and render the PDMS device hydrophobic (longer bake times result in smaller pore sizes within PDMS and improve its hydrophobicity).

## Single emulsion generation

Single emulsions in the comprehensive dataset were generated using a microfluidic device made out of polycarbonate using a low-cost desktop micromill, as previously described[37]. Single emulsions generated here for assessing the accuracy of design automation were made using a double emulsion generator device (Fig. 5a.iv) while blocking the inlet for the outer fluid and essentially only generating droplets at FF1. Similarly, SEs can also be made using a DE generator device by flowing the continuous fluid (here oil) through both middle and outer fluid inlets. The dataset on SE generation is available in an Open Science Framework repository: https://osf.io/938rs/.

## Double emulsion generation

Double emulsions were generated using PDMS microfluidic devices fabricated as described above[32]. A variety of inner and outer fluids commonly used in life science applications were used as inner and outer fluids. HFE 7500 fluorinated oil (Sigma-Aldrich) with a viscosity of 1.6 mPa.s with added 2.2% ionic PEG-Krytox surfactant (FSH 157, Miller-Stephenson) was used as the middle fluid (i.e., oil). The surfactant added to the outer fluid varied depending on its properties (as detailed in Table 1); in all experiments, we added 2% Pluronic F68 with or without added 1% Tween-20, except for the complete RPMI 1640 cell media experiments, where only 5% added Pluronic F127 (Sigma-Aldrich) was used to stabilize the DEs. The majority of DE datapoints used to create a comprehensive dataset and initially train models were taken from our previous study using a single device geometry[32].

We also generated additional data for droplets with a broader range of inner and outer DE diameters to test model prediction accuracy. The additional data were generated using two additional DE generation devices that were either scaled down or scaled up versions of the original device (orifice width at FF1 set to 15 or 30 μm (instead of 22.5 μm) and orifice width at FF2 set to 30 or 60 μm (instead of 45 μm) while keeping normalized channel depth, normalized outlet width, and normalized dispersed and continues fluid inlet widths to 1). Prior to running experiments, each device was surface treated to render the 2nd half of the device (FF2) hydrophilic. This was achieved by taping (Scotch tape) over the inner and middle fluids inlets (to protect the 1st half of the device, FF1, from being exposed to plasma) and allowing air/oxygen plasma to enter through the outer fluid inlet and the device outlet (10 min of plasma treatment)[17,19]. Fluids were introduced to the microfluidic device using syringe pumps (Harvard Apparatus) using 0.015" I.D. and 0.043" O.D. LDPE polyethylene medical tubing (BB31695-PE/2, Scientific Commodities). All fluids were filtered using 0.45 μm polyvinylidene fluoride (PVDF) membrane filters (Millipore) before loading. Within few minutes after surface treatment, we flowed the outer fluid (aqueous sheath) into the devices to ensure that the flow-focuser 2 region of the device remained hydrophilic; 30 s after the introduction of the outer fluid, we introduced the middle fluid (oil) and inner fluid. The flow rates of the middle and inner fluids were initialized with a value higher than the intended final value to speed up fluid entry into the flow-focusers and then slowly lowered to the intended flow rates. Once flow rates were set on syringe pumps, we waited 4-min intervals before collecting droplets to ensure flow stability. Droplet generation was imaged using a high-speed camera (ASI174MM, ZWO) mounted on a stereo-microscope (AmScope). Once DEs were collected, they were imaged inside a cell-counter chamber slide (Countess). The dataset on DE generation is available as an Open Science Framework repository: https://osf.io/938rs/.

## Training machine learning models

The comprehensive dataset used to train models is relatively small (~1000 datapoints) compared to datasets traditionally used to train deep neural networks. We therefore used a shallow neural network comprised of two hidden layers of 512 and 16 nodes with rectified linear units (ReLU) activation functions[51]. We trained the model to minimize a mean squared error loss using an Adam optimizer with a learning rate of 0.0003 and batches of size 32[68]. We also L2 regularized the model parameters with a penalty term of 0.001 to prevent the model from overfitting[57]. This model took 7 design parameters as inputs (orifice width plus 6 dimensionless numbers: capillary number, flow rate ratio, normalized channel depth, normalized dispersed fluid inlet width, normalized continuous fluid inlet width, and normalized outlet channel width). The model then predicted a dimensionless droplet diameter (normalized by the hydraulic diameter of the orifice) as the output. We did not include viscosity ratio as an input parameter for the neural network since it resulted in slightly lower accuracy in predicting unseen previously published and additionally generated data, despite resulting in slightly higher accuracy when predicting the comprehensive dataset used for initial training and testing.

We used the XGBoost package for implementing the boosted decision trees[69]. Our model consists of 100 boosted trees, trained to minimize a mean squared error loss with an L2 regularization penalty term of 1 and a learning rate of 0.3. To prevent individual trees from overfitting, we limited tree depth to 6 and halted leaf node splitting once their weight was below 1. This model took 8 design parameters as inputs (orifice width plus 7 dimensionless numbers: viscosity ratio, capillary number, flow rate ratio, normalized channel depth, normalized dispersed fluid inlet width, normalized continuous fluid inlet width, and normalized outlet channel width). The model then predicted a dimensionless droplet diameter (normalized by the hydraulic diameter of the orifice) as the output.

We assessed models by randomly partitioning the comprehensive dataset into train and test sets comprising 80% and 20% of the original dataset, respectively. Table 3 lists the average accuracy metrics against the test set across 15 randomized training sessions. The scatter plots of predictions for all figures depict results from a single representative run. The source codes for training, testing, and validating the neural network and the boosted decision trees are available on Open Science Framework: https://osf.io/938rs/ and our GitHub repository: https://github.com/CIDARLAB/DAFD-website.

## Parameter significance study

For boosted decision trees, we defined parameter significance as the average loss reduction across the node splits where the parameter serves as the decision variable (also referred to as gain in XGBoost). We calculated each parameter's significance for 15 randomized training sessions and report their averaged significance in Fig. 4f.i. We also repeated our evaluation of parameter significance on subsets of the comprehensive dataset, constraining each subset to only include certain datapoint types: single emulsions, double emulsion inner diameters, and double emulsion outer diameters (depicted in Fig. 4f.ii, iii, iv, respectively).

## Single emulsion design automation

In DAFD 3.0, users first select whether to generate a design for single or double emulsions. For single emulsions, the user enters the desired droplet diameter and/or generation rate, rheological properties of fluids (e.g., viscosities, interfacial tension), and any constraints to the geometric design or flow conditions of the droplet generator. A custom iterative optimization algorithm is then used to find a design and flow rates that deliver the desired performance, as described previously[37]. First, the closest experimental datapoint is found that fits the design constraints. If this closest point fits all constraints and produces a diameter and rate within 3 μm and 15 Hz, respectively, the

experimental point is returned without design iterations. If the closest point is not within these ranges, an iterative optimization process is implemented. For a maximum of 5000 iterations, each parameter is stepped up or down by a specific amount, unless this parameter is constrained or its value passes the preset parameter bounds (Supplementary Table 6). In this workflow, the average prediction of both the neural network and boosted decision trees are used to predict droplet diameter. Prediction accuracy is determined by a model error cost function:

$$C(x) = |\tilde{D}_{desired} - \tilde{D}_x| + |\tilde{F}_{desired} - \tilde{F}_x| \tag{14}$$

where $C(x)$ is the cost of a design $x$, $\tilde{D}$ is the scalar normalized droplet diameter, and $\tilde{F}$ is the scalar normalized generation rate. Both droplet diameter and generation rate are scalar normalized to a standard scalar to prevent bias from the larger range of possible generation rates (hundreds to thousands) compared to diameters (tens to few hundred). If the user only specifies diameter or rate, only that value is included in the cost function. This process is repeated until the cost function reaches zero, the maximum number of iterations is reached, or the change in the cost function is less than a preset tolerance of $10^{-9}$. The geometric design parameters and the flow conditions of the final solution is then returned to the user alongside the predicted droplet diameter and generation rate. In addition, the predicted diameters and generation rates for flow rates up to ±25% of the designed value are provided to construct a performance heat map as a device-specific operation guideline for users. The source code for our SE design automation algorithm is available on our GitHub repository: https://github.com/CIDARLAB/DAFD-website.

## Double emulsion design automation

Design automation of double emulsion generators requires pairing two droplet generators in series. First, the user provides the desired inner and outer diameters and the rheological properties of their desired inner, middle, and outer fluids. Next, six preset devices can be selected, with orifice widths of 15, 22.5, and 30 μm at FF1 (30, 45, and 60 μm at FF2, respectively) and a normalized channel depth of 1 or 1.33. If none of the six devices are selected by the user, all are considered in the design automation workflow. The user can also specify a custom double emulsion generator geometry if preferred. After taking the user inputs, the entire flow space of the two droplet generators is simulated (50–650 μL/h for the inner aqueous fluid, 200–1200 μL/h for the middle fluid, and 1500–10,000 μL/h for the outer aqueous fluid). The dispersed flow rate of flow-focuser 2 (FF2) is simulated across all unique combinations of the total flow rate of FF1 (inner aqueous fluid flow rate plus the middle fluid flow rate) to ensure that the final design is compatible with conservation of mass. Each of the datapoints of FF1 is then paired with points from FF2 that have matching flow conditions and less than a 5% predicted generation rate difference (GRD). Any designs outside of a 5% GRD are deemed unstable and excluded from consideration. No solution is returned if no points with a GRD less than 5% are found. The pairings with GRD <5% are then ranked according to the total percentage error in their predicted inner and outer diameters from the user-specified values. Top candidate designs are then recommended to the user, allowing prioritization of designs that return a certain generation rate, or specific inner or outer diameters. The source code for DAFD 3.0 and the design automation workflow are available at https://github.com/CIDARLAB/dafd-website.

## Reporting summary

Further information on research design is available in the Nature Portfolio Reporting Summary linked to this article.

## Data availability

The comprehensive dataset created and curated in this study and its subsets (SE and DE datasets) have been deposited in the Open Science Framework (OSF) repository and are available at https://osf.io/938rs/ and DAFD's website at http://dafdcad.org. Source Data are provided with this paper.

## Code availability

All source code generated and used in this study for performance prediction and design automation SE and DE droplets are available at https://github.com/CIDARLAB/DAFD-website (citable at https://doi.org/10.5281/zenodo.10156784[70]) and the associated data necessary for running the training models are available at https://osf.io/938rs/.

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

## Acknowledgements

A.L. acknowledges funding as a Damon Runyon Postdoctoral Fellow (DRG-2479-22) from Damon Runyon Cancer Research Foundation. D.P.M. acknowledges funding from the Society of Lab Automation and Screening Graduate Education Fellowship. S.G.K.C. acknowledges funding as a ChEM-H CBI fellow and a Stanford Graduate SGF fellow. D.M.D. and D.P.M. are supported by NSF Semiconductor Synthetic Biology for Information Storage and Retrieval (Award #2027045). P.M.F. is a Chan Zuckerberg Biohub investigator and this work was partly supported by NIH DP2 GM123641 awarded to P.M.F., a Stanford Bio-X interdisciplinary seed grant, and the Emerson Collective.

## Author contributions

A.L. established the research idea, created the SE dataset, trained models, designed and carried out experiments, analyzed data, and wrote the manuscript. D.P.M. helped establish the research idea, developed the algorithms for design automation, and incorporated performance prediction and design automation for DAFD website. S.G.K.C. measured all fluid properties and created the DE dataset. K.K. helped with training machine learning models and improved their generalizability. D.M.D. helped establish the research idea, supported the generation of the SE dataset and development of DAFD, and partially funded the project. P.M.F. helped establish the research idea, supported the generation of the DE dataset, helped write the manuscript, and oversaw and funded the project. All authors read and approved this manuscript.

## Competing interests

The authors declare no competing interests.
