## [Peer Review File · Nature Communications]

Reviewers' Comments:

Reviewer #1:

Remarks to the Author:

Review of: "Design Automation of Microfluidic Single and Double Emulsion Droplets with Machine Learning," by Lashkaripour et al.

This paper is description of how machine learning techniques can be used to design microfluidic chips that make drops, both single drops and double drops. The authors analyze several different conditions to collect a data set and show that their machine learning model is better than other methods to correlate the data. They then show that they can use this sort of model to optimize the size of several different devices to produce single or double drops of desired dimensions. The authors claim that their machine learning method is so easy to use that even labs with no experience in microfluidics will be able to improve their devices with this.

This is an interesting paper and should certainly be published. I did not go through all the details of the paper to see if I could see any errors, but I assume the authors have done this. In my view, this paper is a 'how to' manual. It explains how to set up the machine learning technology to optimize the dimensions of prescribed microfluidic devices. While this is certainly valuable, in my experience, there are so many other factors that need to be optimized in general when making emulsions of different sorts including materials choice, wettability, compatibility and interactions of fluids that this prescription, while very helpful, is still not sufficient to fully design a new device for a new set of materials. Moreover, the paper makes the whole process seem very complicated, and I very much doubt that labs that are just users of microfluidics will want to get involved with this. Nevertheless, as a 'how to' paper this should be published.

My problem with this paper is that I just don't see how it fits in Nature Communications. This is ultimately an editorial decision, but Nature Communications is not the journal where I would look for this sort of paper, and I don't see how this is a general interest paper in any way, shape or form. This is a very highly specialized paper meant for the few labs that seriously design microfluidic devices. In addition, this paper is not a paper about new applications of machine learning but is instead about its application to the specific task of microfluidics design; this is very useful for its intended purpose but not as a general paper about machine learning. As such, this paper would be a fantastic contribution to a journal specialized for microfluidics device design. I just don't see Nature Communications as this sort of journal.

I cannot recommend this paper, nor any revised version of it, for publication in Nature Communications.

Reviewer #2:

Remarks to the Author:

Author presented work in using trained machine learning models to predict droplet diameter and generation rates. This work is exciting since the use of such tool can help shortening the typical process of device design and flow rate exploration. However, after reading through the manuscript and SI, I think there's still work to do to achieve its true potential.

1. In 2.1 only polycarbonate devices were covered for SE and only PDMS for DE drops. What about glass devices, since author initially mentioned device material to be part of the study in the previous paragraph? Quoting from 2.2.2 "we trained a neural network that takes capillary number, flow rate ratio, and five geometric parameters (orifice width, normalized channel depth, normalized outlet width, normalized dispersed fluid inlet width, and normalized continuous fluid inlet width) as inputs and predicts normalized droplet diameter." No material diversity and surface properties (e.g. hydrophobicity/hydrophilicity) was covered in the ML model. Device material and properties are critical in actual experiments and could be the key difference compared to prior art, as author said "previously proposed models only account for variations in flow rates and device geometries [35] or surfactants [36]". The device material variation was ignored or overlooked in the later sections of the manuscript.

2. In 2.1.2, for DE generation, orifice at FF2 is always twice as wide and deep as orifice at FF1. This seems a very constrained device geometry. Can author show how the same ML be used for device that doesn't have this 2x constraint? In section 2.4, author tried to demonstrate using a new DE device with as-yet-unseen channel geometries. But the design variation was not enough to cover all common variations.

3. In paragraph describing Fig 4a, the rate at FF1 and FF2 was used as the method to determine stable and unstable DE generation. This is a fairly constrained requirement and the use of this is reflected in the model's poorer prediction when it comes to DE. In addition, there are commercially-available and custom-made DE devices when the three fluids are meeting almost at the same time for DE production, constrains described above won't be applicable to those cases.

4. Droplets from devices can be much smaller than 25 μm and much bigger than 250 μm . Can author comment on this? This could significantly increase the impact of the tool.

5. Figure 4e. v and related paragraph, from an end application perspective, I would argue that the ability to predict stable DE generation would be much more important than predicting the mode of instability. As this ML model failed to do so, I found it less useful. Author mentioned "This prediction performance could likely be improved in the future by training on a dataset including a much larger number of unstable generation cases." I would like to see that improvement.

6. In the manuscript, author should use $\text{mPa}\cdot\text{s}$ instead of mPa.s , same for similar unit formats in table 2 too. In section 2.2.1, author wrote 58.7-3023%.

Reviewer #3:

Remarks to the Author:

Authors present a microfluidic design automation method for microfluidic flow-focusing generation of single and double emulsion droplets based on a user-friendly online open-source tool developed from machine learning results. This manuscript can be valuable in respect to stable generation of double emulsion having single core because many parameters become narrow compared to generation of monodisperse single emulsion. Overall, the manuscript can provide a very useful tool for many researcher and industrial users. Authors need to be addressed in detail prior to its acceptance to the journal as follows;

1. In my opinion, this study seems to have no new approach, just an extension from the previously developed tool related to single droplet generation (Nature Communications, 2021, 12:25).

2. For single emulsion generation, authors applied the FFG with narrow orifice that have two different widths of collecting channel (Fig. 3a). Meanwhile, the channel geometry for double emulsion generation is a cross channel with simple straight geometry (Fig. 4a). Authors need to clarify why different channel geometries are used to form single and double droplets.

3. Also, even though generalizable models are developed using the FF with narrow orifice, the FFG with simple cross channel is used in Fig. 6a. If FF with simple straight geometry is advantageous for double droplet formation compared to FFG with narrow orifice, comparison of two different geometries for double droplet generation is needed.

4. In this work, two types of polymer are used for base material of microfluidic device, polycarbonate for single emulsion generation and PDMS for double emulsion. Although parameters including flow rate, channel dimension, and capillary number are the same in two types of device, the emulsion generation conditions might be not the same due to different wettability between channel surface and liquid. I think it should be considered.

Response to Editors:

We were encouraged by the Reviewers' enthusiasm for the manuscript during the first round of review. Reviewers 1 and 2 commented that "this is an interesting paper and should certainly be published... and as a how to paper, this should be published" and "this work is exciting since the use of such tool can help shortening the typical process of device design and flow rate exploration."; Reviewer 3 stated "the manuscript can provide a very useful tool for many researcher and industrial users." Thus, reviewers clearly recognize the importance of machine learning approaches in streamlining the design process to enable widespread adoption of droplet microfluidics for life science applications.

We appreciate the Reviewers' thoughtful comments and clear suggestions. In response, we performed extensive additional experiments and analyses that we feel have considerably strengthened the paper:

1. Several Reviewers were concerned that our machine learning models may not generalize to devices fabricated from materials other than PDMS and polycarbonate. To address these concerns, we added new data demonstrating accurate prediction of diameters and generation rates for droplets made using glass devices with an as-yet-unseen fluid combination and device geometry.
2. Reviewer 2 requested additional data detailing whether our machine learning models can accurately predict if flow rate combinations will drive stable or unstable DE generation. To address this, we generated droplets across a variety of unstable regimes, tested the accuracy of our models on these previously unseen data, and included additional analysis assessing accuracy as a function of the allowable difference between droplet generation rates at flow focuser 1 and flow focuser 2.
3. Reviewer 3 raised a concern about whether the current work represents a sufficiently significant advance relative to the previously published version (*Nature Communications* 2021). To address this, we have added a supplementary table highlighting major differences between the 2 publications (e.g. DAFD 3.0 uniquely predicts droplet diameters and generation rates for a wide variety of biologically relevant fluids, enables design automation for double emulsion droplets, and leverages new model architectures to dramatically improve prediction accuracy).

We appreciate the opportunity to submit a revised manuscript that **includes multiple additional experiments and analyses, an additional 5 supplementary figures, and significant edits to the main text and supplementary information.** We hope this new material fully addresses any concerns and that this work can now be considered for publication at *Nature Communications*.

Response to Reviewers:

We appreciate the Reviewers' time and helpful suggestions feedback in reviewing our manuscript. In response to their suggestions, we have performed multiple new experiments and analyses and feel these changes have improved the manuscript and its suitability for the broad audience of *Nature Communications*. Below, we address the Reviewers' concerns point-by-point.

Reviewer 1:

1.1 This paper is description of how machine learning techniques can be used to design microfluidic chips that make drops, both single drops and double drops. The authors analyze several different conditions to collect a data set and show that their machine learning model is better than other methods to correlate the data. They then show that they can use this sort of model to optimize the size of several different devices to produce single or double drops of desired dimensions. The authors claim that their machine learning method is so easy to use that even labs with no experience in microfluidics will be able to improve their devices with this. This is an interesting paper and should certainly be published. I did not go through all the details of the paper to see if I could see any errors, but I assume the authors have done this. In my view, this paper is a 'how to' manual. It explains how to set up the machine learning technology to optimize the dimensions of prescribed microfluidic devices. While this is certainly valuable, in my experience, there are so many other factors that need to be optimized in general when making emulsions of different sorts including materials choice, wettability, compatibility and interactions of fluids that this prescription, while very helpful, is still not sufficient to fully design a new device for a new set of materials.

We appreciate the Reviewer's kind words that *'this is an interesting paper that should certainly be published'*! The Reviewer raises a concern (shared by other Reviewers) that the design algorithm presented here will not be sufficient to fully design a new device for a new set of materials, as factors such as material choice, wettability, and compatibility and interactions between fluids can dramatically impact emulsion formation. In response, we used the previously trained model to attempt to predict droplet size and generation for a previously unseen dataset of droplets comprised of 52% glycerol water in silicone oil (an as-yet-unseen fluid and surfactant combination)⁴ produced using a commercial glass microfluidic device (from Dolomite) with a 195 x 190 μm semi-circular cross-section (**Figs. 5a.ii & R1**). In this dataset, droplets were generated using either 50 mM dodecyltrimethylammonium bromide (C12TAB) surfactant, 5 mM hexadecyltrimethylammonium bromide (C16TAB) surfactant, or no surfactant each at 7 different flow rates (21 total datapoints). For this new dataset, our machine learning models still accurately predicts droplet diameters and generation rates (MAPEs of 17%, 13.5%, and 14.7% for the neural network, boosted decisions trees, and consensus models, respectively) (**Figs. 5b & R2**). Some droplet diameters in this dataset are as large as 420 μm , well outside the range of data used in training. If we limit the data to the training range of our models (diameters of 250 μm or smaller) our models become even more accurate (MAPEs of 15.4%, 11.6%, and 13.4% for the neural network, boosted decisions trees, and consensus models, respectively). This is a rigorous test of the generalizability of our models, given that droplets are generated using a glass device that has a non-rectangular cross-section, a completely new silicone oil (instead of mineral oil or HFE 7500 fluorinated oil in our training set), and a previously unseen dispersed fluid (52% glycerol in DI) with 3 different surfactant conditions.

Fig. R1. Schematic showing device geometry, surface properties, and fluid compositions of a commercial glass device used to generate SE droplets with previously unseen fluids.

These results are also consistent with prior observations that as long as surface properties are favorable for droplet generation (i.e. continuous fluid completely wets the channel, which is determined by a threshold contact angle of $\sim 92^\circ$), diameter and generation rate are independent of surface properties and device material.¹⁻³

Overall, we have made the following changes to the manuscript:

Introduction: We added “Additionally, we demonstrate that our models generalize to new device geometries, fluids, and materials by experimentally validating ‘blind’ predictions.”

Results Section 2.1.2: We added “As long as surface properties favor droplet generation (i.e. the continuous fluid completely wets the channel), droplet diameter and generation rate are independent of surface properties or device material¹⁻³. Therefore, we do not include device material or surface properties as design inputs.”

Figure 5: Figure 5 was redrawn to accommodate the new dataset on droplet generation using glass devices.

Figure S18: Supplementary Fig. 18 was added to compare scaling law and machine

Fig. R2. Comparison between observed diameters (generated using a glass device and unseen fluids) and those predicted by a previously proposed scaling law and 3 different trained models for a single representative training session.

learning models predictions with experimentally calculated generation rates for the previously published and newly generated data.

Results Section 2.4: This section was rewritten to accommodate the new dataset on using a glass microfluidic device to generate droplets.

Table 1: This table was updated to include the different surface properties and device materials used to train our models and assess their generalizability.

1.2 Moreover, the paper makes the whole process seem very complicated, and I very much doubt that labs that are just users of microfluidics will want to get involved with this. Nevertheless, as a ‘how to’ paper this should be published.

The DAFD 3.0 online tool is very user-friendly, requiring only the desired droplet diameter(s) and fluid properties to design a flow-focusing device. Given a particular device design (including commercial microfluidic devices), DAFD 3.0 can also specify optimal flow rates required to make droplets with a given geometry and generation rate. Thus, we hope that this tool proves useful for experts and non-experts alike in making droplet generation less complicated.

1.3 My problem with this paper is that I just don’t see how it fits in Nature Communications. This is ultimately an editorial decision, but Nature Communications is not the journal where I would look for this sort of paper, and I don’t see how this is a general interest paper in any way, shape

or form. This is a very highly specialized paper meant for the few labs that seriously design microfluidic devices. In addition, this paper is not a paper about new applications of machine learning but is instead about its application to the specific task of microfluidics design; this is very useful for its intended purpose but not as a general paper about machine learning. As such, this paper would be a fantastic contribution to a journal specialized for microfluidics device design. I just don't see Nature Communications as this sort of journal. I cannot recommend this paper, nor any revised version of it, for publication in Nature Communications.

We respectfully disagree with the Reviewer. This publication -- while certainly of interest for labs that seriously design microfluidic devices -- is meant for the much broader audience that would like to adopt microfluidics but lacks the expertise to do so. Attesting to the broad interest in efforts like this, we previously published a manuscript describing an earlier and less powerful version of this tool (Lashkaripour *et al. Nature Communications* 2021) that has already been cited >87 times and motivated other efforts to employ machine learning for design automation of droplet generators and even other microfluidic components.⁶⁻¹⁰

The current manuscript represents several important advances over the prior work that will likely lead to even broader adoption (**Table R1**). The prior version was trained only on model fluids and could not generalize to fluids commonly used in bioscience assays unless users provided a small-scale dataset on the new fluid combination and were familiar with machine learning and coding in Python. Here, we trained our model on a new comprehensive dataset that includes a wide variety of geometries, biologically-relevant fluids, and devices and we demonstrate its ability to generalize still-further by experimentally testing 'blind' predictions involving novel device designs and as-yet-unseen fluids and geometries. Additionally, we have extended our design automation and performance prediction capabilities to double emulsion droplets while increasing the maximum generation rate to 12000 Hz (from 500 Hz).

Table R1. Comparison of design capabilities and parameter range of the tool presented in the current submission with DAFD 1.0 (*Nature Communications* 2021)

Metric	Current submission	2021 publication
Droplet size/rate prediction	✓	✓
Droplet diameter range	15 - 250 μm	25 - 250 μm
Droplet generation rate range	5 - 12000 Hz	5 - 500 Hz
Diameter prediction error	5.4%	11.3%
Generation rate prediction error	16.6%	33.5%
As-yet-unseen fluids and geometries prediction error	10.1%	92.7%
Range of interfacial tension	0.31 – 29 mN/m	5 mN/m
Range of dispersed fluid viscosity	0.86 – 42 mPa·s	1 mPa·s
Range of continuous fluid viscosity	1.3 – 57 mPa·s	57 mPa·s
Oil-in-aqueous droplets	✓	
Double emulsions size & rate prediction	✓	

Fluid agnostic design automation	✓	
Double emulsion design automation	✓	
Predicts dimensionless droplet diameter	✓	

In response to the Reviewer comments, we now highlight the major improvements of the current submission compared to our 2021 publication in *Nature Communications* as follows:

Discussion: We added: DAFD 3.0's diverse training set and an improved training approach allowed for it to significantly outperform DAFD 1.0 in terms of accuracy, range of design parameters, and generalizability to as-yet-unseen fluids and geometries (Supplementary Note 4, Supplementary Table 7, and Supplementary Figure 21 & 22).

Supplementary Note 4: We added a new supplementary note, "Comparison of DAFD 3.0 and DAFD 1.0 in generalizability to as-yet-unseen fluids and geometries", that details improvements of DAFD 3.0 over DAFD 1.0.

Supplementary Table 7: We added a new supplementary table (Table R1) that compares this work (DAFD 3.0) and our 2021 publication in terms of parameter range, accuracy, and features enabled by new algorithms.

Supplementary Figures 21 & 22: We added Supplementary Figure 21 that compares the generalizability of the current tool (DAFD 3.0) with our 2021 publication (~10% error for DAFD 3.0 vs ~90% error for the 2021 tool). In Supplementary Figure 22, we demonstrate this improvement in generalizability is partly because of the diversity of the new comprehensive dataset and partly because of our improved training approach (dimensionless output, L2 regularization, and shallow yet wide neural network structure).

Reviewer 2:

Author presented work in using trained machine learning models to predict droplet diameter and generation rates. This work is exciting since the use of such tool can help shortening the typical process of device design and flow rate exploration. However, after reading through the manuscript and SI, I think there's still work to do to achieve its true potential.

We highly appreciate the reviewer's kind words that *'the use of such a tool can help shorten the typical process of device design and flow rate exploration'*! We also appreciate the Reviewer's suggestions on how to improve the manuscript to achieve its true potential. In response, we have made substantial changes (as detailed below).

2.1 In 2.1 only polycarbonate devices were covered for SE and only PDMS for DE drops. What about glass devices, since author initially mentioned device material to be part of the study in the previous paragraph? Quoting from 2.2.2 "we trained a neural network that takes capillary number, flow rate ratio, and five geometric parameters (orifice width, normalized channel depth, normalized outlet width, normalized dispersed fluid inlet width, and normalized continuous fluid inlet width) as inputs and predicts normalized droplet diameter." No material diversity and surface properties (e.g. hydrophobicity/hydrophilicity) was covered in the ML model. Device material and properties are critical in actual experiments and could be the key difference compared to prior art, as author said "previously proposed models only account for variations in flow rates and device geometries [35] or surfactants [36]". The device material variation was ignored or overlooked in the later sections of the manuscript.

Reviewer #2 brings up a concern (shared by Reviewer #1 in comment R1.1) that the datasets used to train the model include limited material types and surface properties such that the model may fail to generalize to materials beyond polycarbonate devices for SE droplets and hydrophilic and hydrophobic PDMS for DE droplets. The Reviewer specifically brings up the concern that the model may perform poorly for glass devices (which are already commercially available).

As described in detail in the response to comment R1.1, we directly tested the ability of the trained model to generalize by attempting to predict droplet size and generation rates for a previously unseen dataset of droplets comprised of 52% glycerol water in silicone oil (an as-yet-unseen fluid and surfactant combination)⁴ produced using a commercial glass microfluidic device (from Dolomite) with a 195 x 190 μm semi-circular cross-section (Figs. 5.ii & R1). In this dataset, droplets were generated using either 50 mM dodecyltrimethylammonium bromide (C12TAB) surfactant, 5 mM hexadecyltrimethylammonium bromide (C16TAB) surfactant, or no surfactant each at 7 different flow rates (21 total datapoints). Despite never having seen glass devices before, the trained models accurately predicted droplet diameters (MAPEs of 17%, 13.5%, and 14.7% for the neural network, boosted decisions trees, and consensus models, respectively) (Figs. 5b & R2). If we limit the data to the training range of our models (diameters of 250 μm or smaller), our models become even more accurate (MAPEs of 15.4%, 11.6%, and 13.4% for the neural network, boosted decisions trees, and consensus models, respectively). As detailed in the response to comment R1.1, this ability to generalize validates our choice to limit model inputs to quantities describing device geometries and fluid properties without a need to explicitly consider surface coatings or device materials. These results are also consistent with prior observations that as long as surface properties are favorable for droplet generation (i.e. continuous fluid completely wets

the channel, which is determined by a threshold contact angle of $\sim 92^\circ$), diameter and generation rate are independent of surface properties and device material.¹⁻³

Fig. R2. Comparison between observed diameters (generated using a glass device and unseen fluids) and those predicted by a previously proposed scaling law and 3 different trained models for a single representative training session.

2.2 In 2.1.2, for DE generation, orifice at FF2 is always twice as wide and deep as orifice at FF1. This seems a very constrained device geometry. Can author show how the same ML be used for device that doesn't have this 2x constraint? In section 2.4, author tried to demonstrate using a new DE device with as-yet-unseen channel geometries. But the design variation was not enough to cover all common variations.

To address the Reviewer's concern, we tested the ability of the previously trained models to predict droplet geometries and generation rates for a set of new DE devices with as-yet-unseen and larger channel dimensions with different relative geometries at FF1 and FF2 (Galogahi *et al.*, *Microfluidics & Nanofluidics*; 2021)¹³. While both devices preserved the same 1:2 relationship between FF1 and FF2 widths (either 50 or 100 μm at FF1 and 100 or 200 μm at FF2), the channel depth remained 100 μm throughout both devices to yield 2-fold differences in the ratio of cross-sectional areas at FF1 and FF2 (**Figs. 5a.iii & R3**). Within these devices, the authors generated DE droplets using an as-yet-unseen fluid combination of HFE7500 - trimethylolpropane trimethacrylate (TMPTMA) - 50% glycerol in water, yielding droplets with inner and outer diameters that varied from 74.4 to 139.2 μm and 109.4 to 173.6 μm , respectively. Despite this new variation in the relationship between FF1 and FF2 geometries, our machine learning models remained able to accurately predict droplet diameters (MAPEs of 7.8%, 8.9%, and 7.0% for the neural network, boosted decision trees, and consensus model, respectively (**Fig. R4**). This strong performance likely results because: (1) our models treat FF1 and FF2 independently without requiring any relationship between them, and (2) having FF2 to be 2x FF1 (a non-planar design) facilitates DE generation even when the middle fluid wets the outer channels at FF2 favorably.¹¹ Additionally, FF2 being 2x FF1 allows for DE generation to mainly occur within the dripping-dripping regime, which in turn enables a better control over DE inner and outer diameters.¹²

Fig. R4. Comparison between observed inner and outer diameters (generated using DE devices with same channel depth at FF1 and FF2 and unseen fluids) and those predicted by a previously proposed scaling law and 3 different trained models for a single representative training session.

In response to the Reviewer comments, we made the following changes to the manuscript:

Results, Section 2.1.2: “A serpentine flow resistor between FF1 and FF2 largely decoupled the generation dynamics of FF1 and FF2 and enabled easy surface modification; the orifice at FF2 was twice as wide and deep as the orifice at FF1 to ensure mostly dripping-dripping mode of DE generation for higher chances of stable DE generation and better control over inner and outer diameters.^{11,12,14}”

Results Section 2.4: This section was edited to accommodate the new data on DE outer diameters from Galogahi et al, which used a DE generator device that had the same depth at FF1 and FF2 (and therefore different ratios of cross-sectional area).

Figure 5 & Sup. Fig. 18: Figure 5 was redrawn and Sup. Fig. 18 was added to accommodate generation rate prediction for the new dataset for outer diameter of DEs generated using a device that has the same channel depth (and thus different cross-sectional ratios) at FF1 and FF2.

2.3 In paragraph describing Fig 4a, the rate at FF1 and FF2 was used as the method to determine stable and unstable DE generation. This is a fairly constrained requirement and the use of this is reflected in the model’s poorer prediction when it comes to DE. In addition, there are commercially-available and custom-made DE devices when the three fluids are meeting almost at the same time for DE production, constraints described above won’t be applicable to those cases.

The constraint in generation rate for stable DE generation is independent of generation method: for every outer droplet, a single inner droplet must be generated to create a single-core DE. As the Reviewer points out, however, some methods can couple these two processes such that an outer droplet breakup event trigger an inner droplet breakup to ensure a 1-to-1 rate. As one example, jetting-dripping enables one-step DE generation even with inner fluids that may be harder to emulsify.¹⁴ However, jetting-dripping DE generation methods typically generate inner diameters much larger than the orifice at FF1 and offer little control over the inner diameter. Given that the current tool seeks to provide device geometries and flow rates required to produce droplets with user-defined diameters, we chose here to focus exclusively on DE generation using

two flow-focusers in series, which provide superior control over the resulting droplet inner and outer diameters. Moreover, this approach allowed us to model DE generation as two independent SE generation events, making it possible to train a single model that covers generation of SEs, DEs, aqueous-in-oil, and oil-in-aqueous droplets.

Specifically, we made the following changes to our manuscript:

Results, Section 2.1.2: *“A serpentine flow resistor between FF1 and FF2 largely decoupled the generation dynamics of FF1 and FF2 and enabled easy surface modification; the orifice at FF2 was twice as wide and deep as the orifice at FF1 to ensure mostly dripping-dripping mode of DE generation for higher chances of stable DE generation and better control over inner and outer diameters.”^{11,12,14}*

2.4 Droplets from devices can be much smaller than 25 μm and much bigger than 250 μm . Can author comment on this? This could significantly increase the impact of the tool.

The current version of our tool supports droplet diameters from 15 – 250 μm at generation rates of up to 12000 Hz and is limited only by the range of training data and its sparsity. While our tool is capable of predicting droplet geometries for droplets > 250 μm , predictions outside the range of training data may be less accurate.¹⁹ While comparisons between model predictions and actual measurements for large (125-420 μm) droplets generated using a glass device with non-rectangular orifice and an as-yet-unseen fluid combination (52% glycerol in silicone oil) established that predictions remain fairly accurate, this accuracy drops with droplets > 300 μm (**Figs. 5b & R2**). After testing the ability of the pre-trained models to generalize to this previously unseen large droplet data, for the online tool we incorporated the data into the training set to improve future predictions (and will continue to do so as new datasets are published).

To address the Reviewer’s comment, we edited the manuscript to read:

Results 2.5: *“Finally, after validating DAFD 3.0’s ability in performance prediction and design automation of SE and DE droplet generation with as-yet-unseen fluids, we trained the machine learning models on all data points generated and used here (including previously published and newly generated data used for assessing generalizability) and updated the online tool with the new models.”*

2.5 Figure 4e. v and related paragraph, from an end application perspective, I would argue that the ability to predict stable DE generation would be much more important than predicting the mode of instability. As this ML model failed to do so, I found it less useful. Author mentioned “This prediction performance could likely be improved in the future by training on a dataset including a much larger number of unstable generation cases.” I would like to see that improvement.

We thank the Reviewer for their careful reading and agree that the ability to predict stable droplet generation is extremely important. To include new training data and improve prediction accuracy, we generated droplets under 32 new flow conditions. Of these, 20 yielded unstable droplets, resulting in a final dataset that included 37 unstable conditions (19 producing multiple-core DEs and 18 producing DEs with missing cores) (**Figs. S16 & R5**). These new data were collected by either keeping the inner fluid flow rate constant (at 200 $\mu\text{L}/\text{hour}$) and varying the middle and outer fluid flow rates (between 250–650 $\mu\text{L}/\text{hour}$ and 3000–7000 $\mu\text{L}/\text{hour}$, respectively) or by keeping the inner fluid flow rate constantly low (at either 60 or 80 $\mu\text{L}/\text{hour}$) and varying the middle and outer flow rates (between 60-400 $\mu\text{L}/\text{hour}$ and 3000-7500 $\mu\text{L}/\text{hour}$, respectively).

Fig. R5. Images for unstable DE droplets (multiple & missing cores) generated using 37 new conditions (5 different fluid combinations and varied inner, middle, and outer fluid flow rates).

To assess whether including more stable training data enhances prediction accuracy for unstable data, we trained the models on the entire original comprehensive dataset (instead of a randomly selected 80%) and quantified the accuracy for 'blind' predictions of whether conditions within this newly augmented dataset yielded stable DEs. The inclusion of additional data (both stable training data and unstable testing data) increased the prediction accuracy of neural network, boosted decision trees, and consensus models to 61.3%, 39.2%, and 51.1%, respectively (a notable improvement over the previous accuracies of 50.6%, 23.5%, and 29.4%) (Figs. 4d).

Decreasing the allowable generation rate difference (GRD) between FF1 and FF2 from 20% to 10% led to a nearly 2-fold increase in true negative rate (i.e. ability of models in correctly predicting unstable DE generation) (**Figs. 4d.v**). With the inclusion of new data, our models remained highly accurate in predicting the mode of instability for the 37 unstable datapoints, with the neural network, boosted decision tree, and the consensus model achieving 85.4%, 71.4%, and 80.0% (similar to previous accuracies of 83.5%, 74.1%, and 85.8%) (**Figs. 4d.iv**).

As pointed out by the reviewer the ability to avoid unstable regimes of DE generation is critically important for successful design automation of DEs. We can change the performance of our models in predicting stable and unstable DE generation by vaying GRD thresholds for stability/instability (**Figs. 4b.iv, 4d.v & R6**). For instance, with a GRD threshold of 12.5% the neural network, boosted decision tree, and consensus models correctly predicted stable DE generation for 77.2%, 68.7%, and 77.4%, respectively. The same threshold for unstable data resulted in correctly predicting unstable DE generation for 66.2%, 41.6%, and 57.0%, for the neural network, boosted decision tress, and consensus models, respectively.

Fig. R6. Effect of varying the GRD threshold on true positive rate (correctly predicting stable DE generation) and true negative rate (correctly predicting unstable DE generation).

We also reported the dependence of our model predictions on GRD threshold using receiver operating characteristic (ROC) curves (**Figs. S17 & R7**). In our online DE design automation tool, we set the GRD threshold for stability/instability to a stringent 5% to minimize false positive rate (i.e. the chances of suggesting a flow rate combination that results in unstable DE generation).

Fig. R7. Receiver operating characteristics (ROC) curves for predicting stable and unstable DE generation while varying GRD threshold for 10 consecutive training sessions. A representative model (median values) is depicted to highlight the effect of GRD on true positive rate (correctly predicting stable DE generation) and false positive rate (falsely predicting unstable DE generation to be stable).

Specifically, we have made the following changes to our manuscript:

Figure 4: Panel “D” of this figure was redrawn to reflect the changes in prediction accuracies with the addition of new unstable DE data. Subpanel “b.iv” was also updated to depict the changes in our ability to correctly predict stable DE generation for different GRD thresholds ranging from 10% to 20%.

Supplementary Figure 16: We added a new supp. Fig. depicting all 37 unstable DE generation datapoints and their respective fluid and flow rate combinations.

Supplementary Figure 17: We added new supp. Fig. that depicts the receiver operating characteristic (ROC) curves for the machine learning models. We also use representative models (median values) to outline the effect of GRD on true positive rate and false positive rate. For all models a GRD 5% represents minimal false positive rates (falsely predicting unstable DE generation to be stable).

Results 2.3: The unstable DE generation part of this section has been updated to reflect the new unstable DE generation dataset. Additionally, the effect of changing the GRD threshold on accuracies of prediction for both stable and unstable DE generation is included and discussed (Figure 4b.iv and 4d.v). We also rewrote the concluding remarks as: “Varying the GRD threshold for stability/instability can help maximize chances of finding a flow rate combination that yield stable DEs or minimize chances of operating at flow conditions that yield unstable DEs (Supplementary Fig.17). The prediction performance of this classification could likely be improved in the future either by training a regime classifier on a dataset that includes an equal representation of stable and unstable DE generation cases.”

2.6 In the manuscript, author should use mPa·s instead of mPa.s, same for similar unit formats in table 2 too. In section 2.2.1, author wrote 58.7-3023%.

We thank the Reviewer for their careful reading of the manuscript. We have corrected the typo in mPa·s throughout the manuscript. For Section 2.2.1, we note that 3023% was the maximum average percentage error for one of the scaling laws we evaluated (Liu *et al.* ²⁰) and is not a typo (Fig. S4).

Reviewer 3:

Authors present a microfluidic design automation method for microfluidic flow-focusing generation of single and double emulsion droplets based on a user-friendly online open-source tool developed from machine learning results. This manuscript can be valuable in respect to stable generation of double emulsion having single core because many parameters become narrow compared to generation of monodisperse single emulsion. Overall, the manuscript can provide a very useful tool for many researcher and industrial users. Authors need to be addressed in detail prior to its acceptance to the journal as follows.

We thank the Reviewer for their positive remarks and insightful suggestions. We have made substantial changes to the manuscript in response to their feedback and feel the paper has been significantly improved as a result.

3.1 In my opinion, this study seems to have no new approach, just an extension from the previously developed tool related to single droplet generation (Nature Communications, 2021, 12:25).

While the current manuscript builds on the prior publication, the current manuscript represents several, substantial improvements:

Data:

- **Extensive new data:** Our 2021 manuscript included 474 datapoints in the dripping regime for SE droplets made using a single and simple fluid combination (DI water and mineral oil). The new dataset includes 868 datapoint on droplet generation in the dripping regime with 11 different fluid combinations. Previously, our dataset was only limited to rapidly prototyped polycarbonate devices with minimum feature sizes (orifice width) of 75 μm . In the new dataset PDMS devices with a minimum feature size of 15 μm were included. Additionally, the new dataset is extended to both aqueous-in-oil and oil-in-aqueous droplets. Finally, the generation rate in the new dataset spans 3 orders of magnitude (5 – 12000 Hz, as opposed to 5 – 500 Hz) (**Table S7**, reproduced here as **Table R1**).

Approach:

- **Generalizability to new fluids and geometries:** Our 2021 DAFD 1.0 model only predicted diameters and generation rates for droplets comprised of DI water in mineral oil; previously extending these predictions to more biologically-relevant fluids required generating a new dataset and using transfer learning (which also required expertise in coding and machine learning). The DAFD 3.0 model presented here can be used with a wide variety of biologically-relevant fluids and can be extended to *any* fluid combination as long as users input fluid properties. Moreover, DAFD 3.0 *dramatically* outperforms DAFD 1.0, increasing utility to users. In Section “2.4 Machine learning models can generalize to previously unseen geometries and fluids” (**Figs. S21 & R8**), we establish that DAFD 3.0 improves the ability to predict droplet diameters with a mean absolute percentage error (MAPE) of 12.1%, 10.7%, and 10.1% for neural networks, boosted decision trees, and consensus models, respectively, from a MAPE of 92.7% for DAFD 1.0.

Fig. R8 Comparison of the generalizability of DAFD 1.0 (Nature Communications 2021) and DAFD 3.0 (current tool) to as-yet-unseen fluids and geometries.

- Improved model training approach:** We further establish that this improvement is not just due to a larger training dataset but results from improvements to the model structure (e.g. use of normalized output, L2 regularization, and a novel neural network structure) (Figs. S22 & R9). While training DAFD 1.0 on the new 2023 comprehensive dataset improves prediction accuracy, the accuracy remains far lower than that for DAFD 3.0 (MAPE of 23.9% vs 12.1%).

Fig. R9. Comparison of the generalizability of DAFD 1.0's neural network trained on SE dataset (2021), DAFD 1.0's neural network trained on the comprehensive dataset, and DAFD 3.0's neural network trained on the comprehensive dataset to as-yet-unseen fluids and geometries.

- Design automation of double emulsions:** DAFD 3.0 newly includes the ability to design devices for double emulsion generation (as discussed in Methods 4.9 Double emulsion design automation).

To clarify these advances, we now include a new Supplementary Table comparing the 2 manuscripts (**Table S7**, reproduced here as **Table R1**):

Overall, we have made the following changes to our manuscript to highlight the major improvements of DAFD 3.0 relative to the DAFD 1.0 model presented in our 2021 publication in *Nature Communications*:

Supplementary Note 4: We include a new supplementary note, “Comparison of DAFD 3.0 and DAFD 1.0 in generalizability to as-yet-unseen fluids and geometries”, that details the improved ability of DAFD 3.0 to generalize over DAFD 1.0.

Supplementary Table 7: We added a new supplementary table (reproduced here as **Table R1**) that compares DAFD 3.0 (this work) and DAFD 1.0 (our 2021 publication) in terms of parameter range, accuracy, and features enabled by new algorithms.

Supplementary Figures 21 & 22: We added Supplementary Figure 21 (which compares generalizability of DAFD 3.0 relative to the DAFD 1.0 model in our 2021 publication) and Supplementary Figure 22 (which establishes that improved performance is not simply due to a larger training set) (reproduced here as **Figs. R8, R9**).

Discussion: We added: “DAFD 3.0’s diverse training set and an improved training approach allowed for it to significantly outperform DAFD 1.0 in terms of accuracy, range of design parameters, and generalizability to as-yet-unseen fluids and geometries (Supplementary Note 4, Supplementary Table 7, and Supplementary Figure 21 & 22).”

3.2 For single emulsion generation, authors applied the FFG with narrow orifice that have two different widths of collecting channel (Fig. 3a). Meanwhile, the channel geometry for double emulsion generation is a cross channel with simple straight geometry (Fig. 4a). Authors need to clarify why different channel geometries are used to form single and double droplets.

Orifice length, defined only when there is an expansion at the outlet channel downstream of the orifice, minimally affects droplet diameter and generation rate in the dripping regime.^{21,22} As a result, we chose not to include the orifice length as an input parameter to our models, allowing us to predict droplet size and generation rate for droplet generators with and without a constriction at the orifice. Validating this choice, our models accurately predicted droplet diameter and rate even for as-yet-unseen device geometries as shown in **Figs. 5a & R10**: panels i & iv (which do not have a restriction), panel iii (which does have a restriction), and panel ii (which features a smooth expansion from the orifice to outlet channel width).

Fig. R10. Schematics showing device geometries, surface properties, and fluid compositions used to generate SE (two left devices) and DE droplets (two right devices) to assess the generalizability of the machine learning models.

To clarify this point in the manuscript, we added the following text:

Results 2.1.2: “As orifice length minimally impacts droplet generation in the dripping regime, we did not consider it as a design parameter²². This enabled us to model flow-focusing geometries with and without an orifice constriction (including a simple straight outlet channel where orifice length cannot be clearly defined).”

3.3 Also, even though generalizable models are developed using the FF with narrow orifice, the FFG with simple cross channel is used in Fig. 6a. If FF with simple straight geometry is advantageous for double droplet formation compared to FFG with narrow orifice, comparison of two different geometries for double droplet generation is needed.

DE droplets are easily generated with or without an orifice constriction. In our revised manuscript, we demonstrate that our model accurately predicts SE and DE droplet diameters and generation rates for as-yet-unseen datasets including droplets generated using devices that do and do not contain orifice constrictions. Devices without orifice constrictions typically maintain the orifice width throughout the outlet channel and require higher pressures to achieve the flow rates required for droplet generation, potentially complicating operation of rapidly prototyped devices with low bonding pressures.²² Apart from these small differences in operating pressure, neither design has an obvious advantage over the other.

3.4 In this work, two types of polymer are used for base material of microfluidic device, polycarbonate for single emulsion generation and PDMS for double emulsion. Although parameters including flow rate, channel dimension, and capillary number are the same in two types of device, the emulsion generation conditions might be not the same due to different wettability between channel surface and liquid. I think it should be considered.

We thank the reviewer for raising this important point on device base material and wettability (a concern shared by other reviewers). As described in detail in the response to comment R1.1, we directly tested the ability of the trained model to generalize by attempting to predict droplet size and generation rates for a previously unseen dataset of droplets comprised of 52% glycerol water in silicone oil (an as-yet-unseen fluid and surfactant combination)⁴ produced using a commercial glass microfluidic device (from Dolomite) with a 195 x 190 μm semi-circular cross-section (**Figs. 5a.ii & R1**). In this dataset, droplets were generated using either 50 mM dodecyltrimethylammonium bromide (C12TAB) surfactant, 5 mM hexadecyltrimethylammonium bromide (C16TAB) surfactant, or no surfactant each at 7 different flow rates (21 total datapoints).

Despite never having seen glass devices before, the trained models accurately predicted droplet diameters (MAPEs of 17%, 13.5%, and 14.7% for the neural network, boosted decisions trees, and consensus models, respectively) (Figs. 5 & R2). If we limit the data to the training range of our models (diameters of 250 μm or smaller), our models become even more accurate (MAPEs of 15.4%, 11.6%, and 13.4% for the neural network, boosted decisions trees, and consensus models, respectively). As detailed in the response to comment R1.1, this ability to generalize validates our choice to limit model inputs to quantities describing device geometries and fluid properties without a need to explicitly consider surface coatings or device materials. These results are also consistent with prior observations that as long as surface properties are favorable for droplet generation (i.e. continuous fluid completely wets the channel, which is determined by a threshold contact angle of $\sim 92^\circ$), diameter and generation rate are independent of surface properties and device material.¹⁻³

References

1. Li, W. *et al.* Screening of the Effect of Surface Energy of Microchannels on Microfluidic Emulsification. *Langmuir* **23**, 8010–8014 (2007).
2. Elvira, K. S., Gielen, F., Tsai, S. S. H. & Nightingale, A. M. Materials and methods for droplet microfluidic device fabrication. *Lab. Chip* **22**, 859–875 (2022).
3. Roberts, C. C. *et al.* Comparison of monodisperse droplet generation in flow-focusing devices with hydrophilic and hydrophobic surfaces. *Lab. Chip* **12**, 1540 (2012).
4. Roumpea, E. *et al.* Experimental studies on droplet formation in a flow-focusing microchannel in the presence of surfactants. *Chem. Eng. Sci.* **195**, 507–518 (2019).
5. Ng, A. H. C. *et al.* MATE-Seq: microfluidic antigen-TCR engagement sequencing. *Lab. Chip* **19**, 3011–3021 (2019).
6. Chagot, L. *et al.* Surfactant-laden droplet size prediction in a flow-focusing microchannel: a data-driven approach. *Lab. Chip* **22**, 3848–3859 (2022).
7. Chen, X. & Lv, H. Intelligent control of nanoparticle synthesis on microfluidic chips with machine learning. *NPG Asia Mater.* **14**, 69 (2022).
8. Barnes, C., Sonwane, A. R., Sonnenschein, E. C. & Del Giudice, F. Machine learning enhanced droplet microfluidics. *Phys. Fluids* **35**, 092003 (2023).
9. Vatandoust, F., Amiri, H. & Mas-hafi, S. DLDNN: Deterministic Lateral Displacement Design Automation by Neural Networks. *ArXiv Prepr.* **2208.14303**, (2022).
10. Talebjedi, B. *et al.* Neural Network-Based Optimization of an Acousto Microfluidic System for Submicron Bioparticle Separation. *Front. Bioeng. Biotechnol.* **10**, 878398 (2022).
11. Rotem, A., Abate, A. R., Utada, A. S., Van Steijn, V. & Weitz, D. A. Drop formation in non-planar microfluidic devices. *Lab. Chip* **12**, 4263 (2012).
12. Calhoun, S. G. K. *et al.* Systematic characterization of effect of flow rates and buffer compositions on double emulsion droplet volumes and stability. *Lab. Chip* 10.1039.D2LC00229A (2022) doi:10.1039/D2LC00229A.
13. Galogahi, F. M., Zhu, Y., An, H. & Nguyen, N.-T. Formation of core-shell droplets for the encapsulation of liquid contents. *Microfluid. Nanofluidics* **25**, 82 (2021).
14. Abate, A. R., Thiele, J. & Weitz, D. A. One-step formation of multiple emulsions in microfluidics. *Lab Chip* **11**, 253–258 (2011).

15. Liu, H., Piper, J. A. & Li, M. Rapid, Simple, and Inexpensive Spatial Patterning of Wettability in Microfluidic Devices for Double Emulsion Generation. *Anal. Chem.* **93**, 10955–10965 (2021).
16. Abate, A. R., Thiele, J., Weinhart, M. & Weitz, D. A. Patterning microfluidic device wettability using flow confinement. *Lab. Chip* **10**, 1774 (2010).
17. Abate, A. R. *et al.* Photoreactive coating for high-contrast spatial patterning of microfluidic device wettability. *Lab. Chip* **8**, 2157 (2008).
18. Romanowsky, M. B. *et al.* Functional patterning of PDMS microfluidic devices using integrated chemo-masks. *Lab. Chip* **10**, 1521 (2010).
19. Lashkaripour, A. *et al.* Machine learning enables design automation of microfluidic flow-focusing droplet generation. *Nat. Commun.* **12**, (2021).
20. Liu, H. & Zhang, Y. Droplet formation in microfluidic cross-junctions. *Phys. Fluids* **23**, 082101 (2011).
21. Lee, W., Walker, L. M. & Anna, S. L. Role of geometry and fluid properties in droplet and thread formation processes in planar flow focusing. *Phys. Fluids* **21**, 032103 (2009).
22. Lashkaripour, A., Rodriguez, C., Ortiz, L. & Densmore, D. Performance tuning of microfluidic flow-focusing droplet generators. *Lab. Chip* **19**, (2019).

Reviewers' Comments:

Reviewer #2:

Remarks to the Author:

I appreciate the author's effort to perform additional experiment and provide supporting data to address the concerns. The latest version has significantly improved over the previous one.

However, I fail to see some of my concerns get fully addressed. I was concerned about the 2x width limit between FF1 and FF2. While the author included another design claiming a 2-fold difference, the width ratio is the same. As author wrote in the response, "having FF2 to be 2x FF1 facilitates DE generation". There are a good amount of work out there that's not constrained to this limit. I truly want to see the model being applied to a wider range of device geometries as this increases its impact for the community. If the model has limitation, then it should be properly described. The current title and writing indicates a more universal suitability which appears untrue based on the data presented.

Reviewer #3:

Remarks to the Author:

This manuscript has been well revised based on the reviewers' comments. I believe that this manuscript will be considered for acceptance into the journal in its current form.

Response to Editors:

We are thrilled that the Reviewers have found the revised version of our manuscript “has significantly improved over the previous one” and “will be considered for acceptance into the journal in its current form”. We were once again encouraged by the Reviewers’ enthusiasm for our revised manuscript during the second round of reviews. We have addressed the remaining concern on universal prediction of double emulsion generation by toning down the claim of universality. We also appreciate the Editors detailed suggestions for improving our manuscript and we have implemented all suggestions within the Authors checklist.

We hope that this revised and improved work can now be considered for publication at *Nature Communications*. We are excited to share this work with the life sciences, microfluidics, and machine learning communities.

Response to Reviewers:

We appreciate the Reviewers’ time and helpful suggestions in reviewing our manuscript in both rounds. In response to the final concern raised by Reviewer 2 we have toned down our claim of universal predictions for double emulsion generation.

Reviewer 2:

I appreciate the author’s effort to perform additional experiment and provide supporting data to address the concerns. The latest version has significantly improved over the previous one.

However, I fail to see some of my concerns get fully addressed. I was concerned about the 2x width limit between FF1 and FF2. While the author included another design claiming a 2-fold difference, the width ratio is the same. As author wrote in the response, “having FF2 to be 2x FF1 facilitates DE generation”. There are a good amount of work out there that’s not constrained to this limit. I truly want to see the model being applied to a wider range of device geometries as this increases its impact for the community. If the model has limitation, then it should be properly described. The current title and writing indicates a more universal suitability which appears untrue based on the data presented.

We are glad that the Reviewer has found the revised version of our manuscript significantly improved. We have appreciated the Reviewer’s detailed comments and insightful suggestions in both rounds of review. Their notable contribution has helped us improve our manuscript notably. For the remaining concern on universal models for DEs we have clarified our generalizability claim to devices with the 2x design constraint.

To address Reviewers comment we have made the following changes:

- To the *Machine learning models can generalize to previously unseen geometries and fluids* section: “This ability to accurately predict data with as-yet-unseen fluids, geometries (**in case of DE generators demonstrated for devices with FF2 orifice widths that are 2-fold the width at FF1**), and device materials demonstrates an ability to generalize, likely due to the diversity of the comprehensive dataset in terms of geometries, fluid properties, and flow rates, the use of dimensionless inputs and output, and L2 regularization during model training.” to the
- To the *Discussion* section: “Here, we establish that machine learning models can enable accurate and generalizable prediction of droplet diameters and generation rates based on device geometries, fluid properties, and flow rates for aqueous-in-oil and oil-in-aqueous SE and DE droplets (**shown for DE generators with FF2 widths 2-fold greater than the width at FF1**).”

Reviewer 3:

This manuscript has been well revised based on the reviewers' comments. I believe that this manuscript will be considered for acceptance into the journal in its current form.

We have appreciated the Reviewer's insightful comments and suggestions during the two rounds of revision, and we acknowledge their contributions to our revised and improved manuscript. We are glad that the Reviewer found the changes to the manuscript satisfactory.